# Koopman Embedded Equivariant Control

## Abstract

An efficient way to control systems with unknown nonlinear dynamics is to find an appropriate embedding or representation for simplified approximation (e.g. linearization), which facilitates system identification and control synthesis. Nevertheless, there has been a lack of embedding methods that can guarantee (i) embedding the dynamical system comprehensively, including the vector fields (ODE form) of the dynamics, and (ii) preserving the consistency of control effect between the original and latent space. To address these challenges, we propose Koopman Embedded Equivariant Control (KEEC) to learn an embedding of the states and vector fields such that a Koopman operator is approximated as the latent dynamics. Due to the Koopman operator's linearity, learning the latent vector fields of the dynamics becomes simply solving linear equations. Thus in KEEC, the analytical form of the greedy control policy, which is dependent on the learned differential information of the dynamics and value function, is also simplified. Meanwhile, KEEC preserves the effectiveness of the control policy in the latent space by preserving the metric in two spaces. Our algorithm achieves superior performances in the experiments conducted on various control domains, including the image-based Pendulum, Lorenz-63 and the wave equation.

## 1 Introduction

Many real-world system dynamics are unknown and highly nonlinear, which limits the applications of classical control methods. Although model-based control methods have been widely studied to learn the dynamics from the data, e.g., in (Chua et al., 2018; Deisenroth et al., 2009; Müller et al., 2012; Nagabandi et al., 2018; Williams et al., 2017), the learned dynamics can still be highly non-linear or black-box, making it still analytically intractable and computationally inefficient. One effective class of methods addressing this issue is to find a proper representation that embeds the dynamical system into a latent space (Ha & Schmidhuber, 2018), in which the system evolution is simple (e.g. locally linear) (Banijamali et al., 2018; Levine et al., 2019; Mauroy & Goncalves, 2016; Mauroy et al., 2020; Watter et al., 2015; Weissenbacher et al., 2022; Williams et al., 2015), such that various control methods such as iterative Linear Quadratic Programming can be used.

However, current embedding methods have primarily focused on next-step predictions through local linearization approaches (Bruder et al., 2019a; Kaiser et al., 2019; Bruder et al., 2019b; Li et al., 2019). The learned dynamics neglect the vector fields or metrics, which lead to the learned latent dynamics and the optimal control policy derived based on it inconsistent with the original ones. These methods lack a formal and theoretical guarantee that the system is comprehensively embedded into the latent space, including the vector fields and flows [1]. Thus, the effect of control may not be preserved when the control policy inversely mapped back to the original space. The sufficient conditions for preserving the control effects are the equivariance of the dynamics (Maslovskaya, 2018) and the metric preservation of the latent space (Jean et al., 2017). In this paper, we aim to find an *isometric* and *equivariant* mapping such that the flows and vector fields of the original nonlinear dynamics are comprehensively mapped to a latent controllable system, thus preserving the control effect in both the original and latent spaces.

Related works in this field fall into two main streams: embedding for control and symmetry in control. Embedding to control algorithms aims to map complex, high-dimensional, nonlinear dynamics into a latent

---

[1] Given the control policy, the system dynamics and trajectories under this control policy can be viewed as vector fields and corresponding flows, respectively (Field, 2007).

space with local linearization, often using Variational Autoencoder (VAE) structures (Watter et al., 2015; Kaiser et al., 2019; Nair et al., 2018). These algorithms ensure a bijection between the original and latent spaces (Huang, 2022; Levine et al., 2019), however, they do not embed the necessary differential and metric information with equivariance, causing the control effect to be inconsistent in both spaces. Conversely, symmetry in control theory has been crucial in identifying invariant properties of systems (Field, 2007), leading to effective control policies. Research has underscored the value of symmetric representations in learning dynamics for tasks with evident (e.g., rotation and translation invariance) symmetries (Adams & Orbanz, 2023; Bloem-Reddy & Teh, 2020; Bronstein et al., 2017), such as work in robotics control tasks under special orthogonal group (SO(2)) action by Wang et al. (2021). However, these methods may struggle in scenarios with unknown dynamics and less evident symmetries. Recent advancements include methods designed to learn implicit symmetries, such as meta-sequential prediction for image prediction with implicit disentanglement frameworks (Miyato et al., 2022; Koyama et al., 2023). These approaches achieve disentanglement as a by-product of training symmetric dynamical systems by Fourier representation. Such properties are important in maintaining consistency in control policies across different spaces.

An additional line of research on embeddings for control involves Koopman operator theory. The Koopman operator was originally proposed by Koopman and von Neumann as a linear embedding for Hamiltonian dynamical systems (Koopman, 1931; Koopman & Neumann, 1932). However, due to its infinite-dimensional nature, identifying suitable handcrafted basis functions is challenging with conventional methods (Brunton et al., 2021). The development of deep learning has greatly advanced this area by enabling data-driven discovery of effective representations of dynamical systems. For instance, extended dynamic mode decomposition (E-DMD) has been introduced as a linear approximation of the Koopman operator (Proctor et al., 2018). Building on this, nonlinear basis functions such as kernel methods and autoencoders have also been explored for system identification in general nonlinear dynamics (Lusch et al., 2018; Liu et al., 2023; Bevanda et al., 2024a;b; 2025; Wu et al., 2025). Thanks to its linearity and universal representational capacity, the Koopman operator has attracted significant attention in unknown nonlinear control tasks. A representative example is the integration of Koopman embedding with model predictive control (Korda & Mezić, 2020). In addition, model-based reinforcement learning (RL) leverages the Koopman operator as an offline model to improve sample efficiency (Weissenbacher et al., 2022). Beyond optimal control tasks, the Koopman operator has also been employed with certificate functions such as Lyapunov functions and control barrier functions to address safety and stability issues (Folkestad et al., 2020; Zinage & Bakolas, 2023).

In this paper, we propose *Koopman Embedded Equivariant Control (KEEC)* to learn an equivariant embedding of dynamical system based on Koopman operator theory. Unlike existing works related to embedding methods and Koopman methods (Chua et al., 2018; Deisenroth et al., 2009; Bruder et al., 2019a;b; Li et al., 2019; Mauroy et al., 2020; Weissenbacher et al., 2022) to merely embed the states, we argue that solely embedding states is insufficient in the latent space. From a geometric perspective, our work addresses a central question in latent embedding: *what constitutes an appropriate embedding for control?* To this end, we formally propose that equivariance and isometry are two essential properties that need be preserved in the latent space to ensure effective control. Based on manifold and Lie theory (Omori, 2017; Lie, 1893), we show that the Koopman operator enables a comprehensive embedding of both equivariant flows and vector fields, thereby preserving the full structure of the dynamics. By incorporating isometry, the invariant value function can be derived, which preserves consistent optimal control performance in both the original and embedded spaces. Leveraging this invariant value function and comprehensive embedding, we further obtain an analytical control policy via Hamiltonian–Jacobi theory, without the need to train a separate policy neural network. This result establishes a convergence toward the optimal value function and enables efficient policy generation. Our numerical experimental results demonstrate KEEC's superiority over existing methods in controlling unknown nonlinear dynamics across various tasks, such as Gym (Towers et al., 2023)/image Pendulum, Lorenz-63, and the wave equation achieving higher control rewards, shorter trajectories and improved computational efficiency. Figure 1 takes pendulum control as an example to demonstrate the control framework of the KEEC and shows the learned vector field on the latent space.

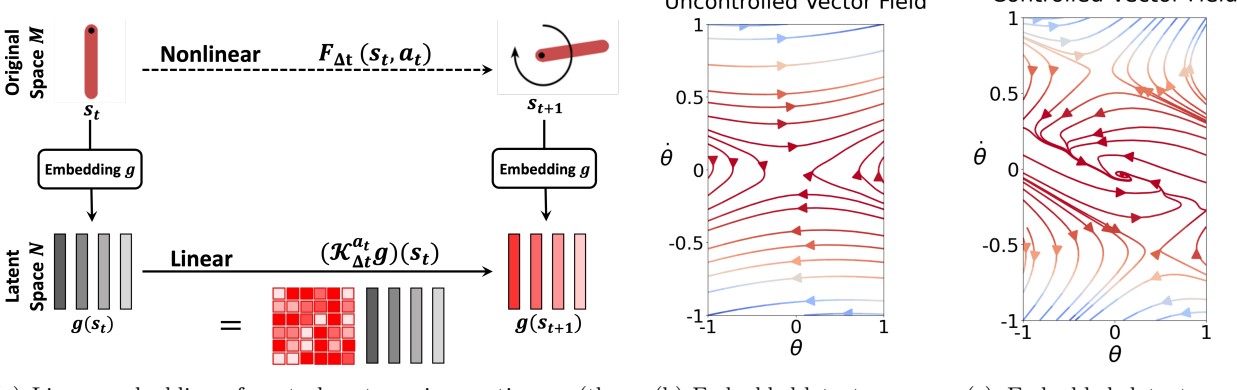

(a) Linear embedding of control system given action $a_t$ (the pendulum trajectory)

(b) Embedded latent uncontrolled vector field

(c) Embedded latent controlled vector field

Figure 1: Figure 1(a) is an overview of KEEC. The left panel (Figure 1(b)) shows the embedded latent uncontrolled vector field of the pendulum, and the right panel (Figure 1(c)) shows the corresponding embedded latent controlled vector field. Under the controlled vector field, the pendulum contracts to the target point. More information is available in Section 4.

## 2 Preliminary

### 2.1 Optimal Control: A Geometric Perspective

State at time $t$, $s_t$ is used to indicate the current status of the system. The collection of all states form state space denoted by $M$. The dynamics in unknown and nonlinear control-affine system is affected by action $a_t$ in action space $A$, usually formulated as an Ordinary Differential Equation (ODE):

$$\dot{s}_t = f(s_t, a_t) = f_M(s_t) + B(s_t)a_t, \tag{1}$$

where $f_M$ and $B$ are called the drift (action-independent) and actuation (action-dependent) terms. The Equation 1 expresses the time-derivative of state $s_t$ given action $a_t$. The collection of time-derivatives at each point $s_t \in M$ forms a **vector field** on $M$.

A **flow** consists of the collection of all trajectories of the states, denoted by $F_{\Delta t}$. $F_{\Delta t} : M \times \mathcal{A} \to M$ is the flow map of the dynamical system, representing the state transition over time $\Delta t$. The system state at time $t + \Delta t$ under control action $a_t$ is:

$$s_{t+\Delta t} = F_{\Delta t}(s_t, a_t) = s_t + \int_t^{t+\Delta t} f(s_\tau, a_\tau)d\tau. \tag{2}$$

The reward function $r : M \times \mathcal{A} \to \mathbb{R}$ is commonly a quadratic form in control system satisfying $r(s, a) = r_1(s) + r_2(a)$. A control policy is a mapping $\pi : M \to \mathcal{A}$. our objective is to maximize the value function

$$V^\pi(s_t) = \mathbb{E}[\sum_\tau \gamma^\tau r(s_\tau, a_\tau) \mid s_\tau, a_\tau \sim \pi], \quad \forall s_t \in M \subset \mathbb{R}^m, \tag{3}$$

where $\gamma \in (0, 1)$ is the discount factor. Other notations are in Appendix A and B.

### 2.2 Embedding for Control

Solving control problems in an unknown and nonlinear system in equation 1 is challenging. Embedding for control is the idea. Thus, we aim to learn an embedding $g$ that transform the system in the original state space $M$ to a latent space $N$, in which the control problem can be easily solved. For example, mapping

the original system to a latent one where the dynamics is linearly evolving. It thus motivates our study by answering the question: *What properties should the embedding g satisfy?*

An embedding $g$ for control problems first should preserve the system dynamics, i.e. mapping the state dynamics from $M$ to $N$. This can be satisfied if the embedding is equivariant.

**Equivariance.** A map $g : M \to N$ is *equivariant function* if $F^{\text{latent}} \circ g = g \circ F$ (Hall & Hall, 2013), where $F^{\text{latent}}$ is the flow in latent space, *equivariant* with respect to flow $F$ under the map $g$.

Equivariant map ensures that $F^{\text{latent}} \circ g(s_t) = g \circ F(s_t)$, i.e., the transitions of states remain consistent between the original and latent spaces. In addition, preserving the metric on both spaces is essential for maintaining the consistency of reward/cost functions and control effects across both spaces.

**Isometry.** Let $M$ and $N$ be metric spaces with metrics $d_M$ and $d_N$. A map $g : M \to N$ is an isometry if for any $s_1, s_2 \in M$, we have $d_M(s_1, s_2) = d_N(g(s_1), g(s_2))$ (Field, 2007).

Isometry can guarantee the reward/cost functions on the original space $M$ and latent space $N$. This is because most reward/cost function is highly dependent on the distance between current state to the optimal state, such as the reward design in MuJoCo (Todorov et al., 2012). Furthermore, the distance-preserving property of the trajectory ensures that the latent and original trajectories are consistent with one another.

An equivariant and isometric map $g$ can comprehensively map the original system to a latent system and guarantee the consistent control performances. And solving the nonlinear control problem with unknown dynamics remains challenging. To simplify the problem, we aim for a latent system that has simple dynamics on the latent space $N$. Works have been done by linearizing the dynamics (Watter et al., 2015; Kaiser et al., 2019). However, they didn't comprehensively map the flows and vector fields original dynamics to the latent space.

## 3 Koopman Embedded Equivariant Control

In this section, we consider an embedding function $g$ mapping the original state dynamics to a latent dynamics. We introduce the Koopman operator for optimal control by proposing **Koopman Embedded Equivariant Control (KEEC)**. We learn an isometric and equivariant function to embed the control dynamics into a latent space such that the latent dynamics can be represented by a Koopman operator. The organization of this section is as follows. We first leverage in section 3.1 the Koopman operator, an infinite-dimensional linear operator, as a equivariant linear representation of dynamical system $F$ in equation 1. Then in section 3.2, in order to embed the vector field $f_M$; we demonstrate that the infinitesimal generator of the Koopman operator rather than the operator itself can be used to represent the equivariant latent vector field. In section 3.3, we demonstrate our methods for learning an equivariant embedding with finite-dimensional approximation of Koopman operator. Finally in section 3.4, we leverage the value-based methods to solve the simplified equivariant control problems.

### 3.1 Koopman Operator as Latent Dynamics

We utilize the Koopman operator to simplify the dynamics of a nonlinear system. We aim to embed the original state space $M$ into a latent space $N$ with function $g : M \to N$. Denote the space of continuously differentiable functions as $C^1(M)$. In an uncontrolled dynamical system, the Koopman operator $\mathcal{K}_{\Delta t} : C^1(M) \to C^1(M)$ is an infinite-dimensional linear operator that governs the evolution of observables (functions in $C^1(M)$) over a time period $\Delta t$ (Das, 2023). Mathematically, for any function $g \in C^1(M)$, the Koopman operator satisfies: $\mathcal{K}_{\Delta t} g = g \circ F_{\Delta t}$.

The inclusion of the control action can be seen as extending the Koopman framework to a system where the evolution of functions depends on both the state and the action. The goal is to define an operator that tracks how functions evolve under the influence of action. In this paper, considering the system evolving as: $s_{t+\Delta t} = F_{\Delta t}(s_t, a_t)$, given action $a_t$, the Koopman operator then describes how the function $g(s_t)$ evolves when an action $a_t$ is applied:

$$g(F_{\Delta t}(s_t, a_t)) = (\mathcal{K}_{\Delta t}^{a_t} g)(s_t). \tag{4}$$

equation 4 demonstrates the equivariance between two dynamics $\mathcal{K}$ and $F$. $\mathcal{K}$ and $F$ essentially describe the same state transition but in different spaces. Formally, $\mathcal{K}^{a_t}(z_t) = z_{t+\Delta t}$, where $z_t := g(s_t) \in N$ and $z_{t+\Delta t} := g(s_{t+\Delta t}) \in N$ are the latent states. Refer to Appendix D for more details about the equivariance.

In the rest of this paper, we omit the superscript action $a$ given the common setting of control. $\mathcal{K}$ pushes the latent state $z$ forward in time along the controlled dynamics. In this way, the map $g$ is an equivariant function and evolves globally in a linear manner due to the Koopman operator, even if the original dynamics is nonlinear. However, merely approximating the Koopman operator $\mathcal{K}$ and map $g$ does not fully capture the vector fields induced by the ODE of the time-derivative dynamics.

### 3.2 Modelling the Equivariant Latent Vector Fields

Vector fields are not embedded with simple Koopman operator since only the dynamics $F$ is mapped in the latent space, and the vector field requiring the derivatives of latent states have been lacking. To address this, we first discuss embedding the drift term in equation 1 by setting no control ($a_t \equiv 0$).

Based on group theory and Lie theory, the Koopman operator $\mathcal{K}$ can be treated as a linear one-parameter semigroup (Bonnet-Weill & Korda, 2024). Given equation 1, $F$ is smooth. Thus, the Koopman operator has an infinitesimal generator $\mathcal{P}$, where $\mathcal{K}_{\Delta t} = \exp(\mathcal{P}\Delta t)$. Here $\mathcal{P}$ is well-defined on the dense subset of function space $C^1(M)$ (more details refer to D).

According to Sophus Lie (Lie, 1893), the one-parameter Lie group is generated by the Lie algebra. In other words, the smooth vector field induces a flow $\mathcal{K}_{\Delta t}$. According to equation 4, the time derivative of the latent states under the flow $\mathcal{K}_{\Delta t}$ is:

$$\mathcal{L}_f g(s_t) = \lim_{\Delta t \to 0} \frac{g \circ F_{\Delta t} - g}{\Delta t}(s_t) = \lim_{\Delta t \to 0} \frac{\mathcal{K}_{\Delta t} g - g}{\Delta t}(s_t) = \mathcal{P}g(s_t). \tag{5}$$

$\mathcal{L}$ denoted as the Lie derivative. equation 5 shows that with no control, the underlying dynamical system can be globally linearized due to that $\dot{g} = \frac{d}{dt}\exp(\mathcal{P}t)g = \mathcal{P}g$. Therefore, the homogeneous part in the ODE, i.e. the drift term in the dynamics vector field in equation 1, can be fully embedded and also linearized.

As for the actuation term of the dynamics, since the homogeneous part is embedded, and the derivatives of states are embedded by map $g$, the remaining inhomogeneous part in the ODE (actuation) is also automatically embedded. Although the actuation term is not linearized, it is still possible in practice to find an embedding $g$ such that the latent actuation term becomes a simple function of latent state, such as constant or linear function. We formally demonstrate the equivariant embedding as follows.

**Theorem 3.1** (Equivariant Vector Field). *Given that the unknown nonlinear control-affine dynamics in equation 1, with the Koopman operator and its infinitesimal generator, the equivariant dynamical system evolves on $N$ as:*

$$\dot{z}_t = \mathcal{P}z_t + (\mathcal{U}z_t)a_t, \tag{6}$$

*where $z_t = g(s_t)$, $s_t \in M$ is the original state, and $\dot{z}_t$ is the derivative w.r.t time $t$. The $\mathcal{U}$ is a state-dependent operator that maps the latent state $z_t$ to a linear operator acting on the action $a_t$. (Further details and Proof in Appendix F.1)*

According to this theorem, the system under Koopman representation has well-defined vector fields equivariant to the original vector fields in the dynamics ODE.

Our embedding enables that the latent dynamics can be simply learned by solving linear equations, which can stabilize the learning procedure (see Section 3.3). Besides, due to the assumption of control-affine systems, the embedded system dynamics actuation part ($\mathcal{U}z_t$) provides the information of vector fields to derive an analytical form of policy extraction, i.e. representing greedy policy with value function (see Section 3.4). We then derive the flow of the equivariant vector field for later embedding learning.

**Proposition 3.2** (Equivariant Flow). *According to derived operators $\mathcal{P}, \mathcal{U}$ in equation 6, the equivariant flow under the two operators can be derived as*

$$z_{\Delta t} \approx \exp(\mathcal{P}\Delta t)z_0 + \mathcal{P}^{-1}(\exp(\mathcal{P}\Delta t) - I)(\mathcal{U}z_0)a_0, \tag{7}$$

*where $I$ is the identity operator. (Proof in Appendix F.1)*

These two theorems describe the differential and integral forms of the latent dynamics and can be derived from one another. The equivariant flow in equation 7 will be used to predict the next latent state, $z_{t+\Delta t} \approx \exp(\mathcal{P}\Delta t)z_t + \mathcal{P}^{-1}(\exp(\mathcal{P}\Delta t) - I)(\mathcal{U}z_t)a_t$.

### 3.3 Learning Equivariant Embedding

To comprehensively embed the original space, two properties need to be satisfied, i.e., equivariance and isometry. Equivariance guarantees that the learned flows and vector fields are consistent under embedding. Isometry makes the metric consistent and preserves the control effect in both spaces. KEEC leverages the auto-encoder structure to learn the equivariant and isometric embedding. And to learn the latent dynamics and its vector field, instead of learning the Koopman operator $\mathcal{K}$ itself, we learn the vector field parameters derived from $\mathcal{K}$: $\mathcal{P}$ and $\mathcal{U}$.

The Koopman operator is inherently infinite-dimensional, and traditional methods that approximate it with finite-dimensional models or manually selected feature functions for embedding (Budišić et al., 2012; Kutz et al., 2016), such as polynomials, often lead to inaccuracies and incomplete representations of nonlinear dynamics. Moreover, these feature functions typically struggle with high-dimensional data, limiting their applicability to large-scale systems (Tu, 2013). Instead, we leverage deep learning, which is well-suited for general-purpose function approximation. In our work, we employ a deep auto-encoder to learn the embeddings that naturally fit with the Koopman framework (Lusch et al., 2018; Brunton et al., 2021). Denote the encoder-decoder pair as $g_\theta^{\mathrm{en}} \colon \mathbb{R}^m \to \mathbb{R}^n, g_\phi^{\mathrm{de}} \colon \mathbb{R}^n \to \mathbb{R}^m$, mapping between original space $M$ and latent space $N$, parameterized by $\theta$ and $\phi$. Let $s_{t_0:t_1}$ represents consecutive states from $t_0$ to $t_1$. The dataset consists of state transitions $\{s^{(j)} = s_{0:L}^{(j)}, s_+^{(j)} = s_{1:L+1}^{(j)}\}_{j=1}^J$ and corresponding actions $\{a^{(j)} = (a_0^{(j)}, ..., a_L^{(j)})\}_{j=1}^J$.

**Identifying the latent dynamics.** Given one tuple of data $(s^{(j)}, a^{(j)}, s_+^{(j)})$, we firstly map the state sequence to the latent space, such as $z^{(j)} = \left(g_\theta^{\mathrm{en}}(s_0^{(j)}), ..., g_\theta^{\mathrm{en}}(s_L^{(j)})\right)$, , and similarly $z_+^{(j)}$ corresponds to $s_+^{(j)}$. We then approximate the $\hat{\mathcal{P}}$ and $\hat{\mathcal{U}}$ by solving a least square problem according to Theorem 3.1 and Proposition 3.2 as

$$\hat{\mathcal{P}}, \hat{\mathcal{U}} = \arg\min_{\mathcal{P}, \mathcal{U}} \|[\exp(\mathcal{P}\Delta t)z + \mathcal{P}^{-1}(\exp(\mathcal{P}\Delta t) - I)(\mathcal{U}z)a] - z_+\|, \tag{8}$$

where $\hat{\mathcal{P}} \in \mathbb{R}^{n \times n}, \hat{\mathcal{U}} \in \mathbb{R}^{n \times d \times n}$. By concatenating $z$ and $z * a$, where $*$ denotes the column-wise Kronecker product, we define $\Lambda = [z, z * a]^T \in \mathbb{R}^{L \times (n+n \times d)}$, the problem in equation 8 can be simplified and yield a solution such as (see equation 28 in Appendix F.1),

$$C = \frac{1}{\Delta t}(z_+ - z)\Lambda(\Lambda^T \Lambda)^\dagger \in \mathbb{R}^{n \times (n+n \times d)}, \qquad [\hat{\mathcal{P}}, \hat{\mathcal{U}}] = [C_{1:n}, C_{(n+1:n+n) \times d}], \tag{9}$$

where $C_{i:j}$ represents the $i$th to $j$th columns of $C$, and we have $\dot{z}_t \approx \hat{\mathcal{P}}z_t + (\hat{\mathcal{U}}z_t)a_t$

**Equivariance Loss.** KEEC learns the equivariant flows and vector fields in the latent space by training the encoder according to our proposed loss function, consisting of two terms. The first is equivariance loss, simplified to the name *forward loss*:

$$\mathcal{E}_{\mathrm{fwd}} = \sum_{t=t_0}^{t_0+(L-1)\Delta t} \left\{ \|g_\phi^{\mathrm{de}}(\hat{z}_{t+\Delta t}) - s_{t+\Delta t}\| + \|g_\phi^{\mathrm{de}} \circ g_\theta^{\mathrm{en}}(s_t) - s_t\| \right\}, \tag{10}$$

where $\hat{z}_{t+\Delta t} \approx \exp\left(\Delta t\hat{\mathcal{P}}\right)z_t + \hat{\mathcal{P}}^{-1}(\exp\left(\hat{\mathcal{P}}\Delta t\right) - I)(\hat{\mathcal{U}}z_t)a_t$. Based on Theorem 3.1 and 3.2, the equivariant flows can be analytically derived by equivariant vector fields. Thus, the $\hat{z}_{t+\Delta t}$ can be explicit via the solutions of $\hat{\mathcal{P}}$ and $\hat{\mathcal{U}}$ from equation 9. This leverages the basic properties of Lie group and algebra in equation 5, and this numerical computation leads a more efficient and robust joint training procedure due to the comprehensive embedding. In such situation, both flows and vector fields can be captured under the embedding $g_\theta^{\mathrm{en}}$.

In the loss function $\mathcal{E}_{\text{fwd}}$, $\|g_\phi^{\text{de}}(\hat{z}_{t+\Delta t}) - s_{t+\Delta t}\|$ represents the correction of the equivariant flow; $\|g_\phi^{\text{de}} \circ g_\theta^{\text{en}}(s_t) - s_t\|$ is the standard identity loss in auto-encoder, imposing the equivariant constraints required by learning the embedding $g_\theta^{\text{en}}$.

Control tasks often rely on metric information (Lewis et al., 2012), which, in our KEEC framework, is implicitly defined on the latent space. The inconsistency in metric information can lead to diverse control effects in the latent space compared to the original space (Jean et al., 2019). A consistent metric by an isometry embedding is a sufficient condition for preserving this control effect.

**Isometry Loss.** Here, we introduce the second loss term *isometry loss*:

$$\mathcal{E}_{\text{met}} = \sum_{t=t_0}^{t_0+(L-1)\Delta t} \left| \|z_{t+\Delta t} - z_t\| - \|s_{t+\Delta t} - s_t\| \right|, \tag{11}$$

which is the absolute error between the distance measured in the latent space and the original space. In fact, $\mathcal{E}_{\text{met}}$ is used to embed metric information consistently. The scale of $\mathcal{E}_{\text{met}}$ evaluates the *Distortion*[2] of latent space induced by embedding $g_\theta^{\text{en}}$. It is worth noting that $\mathcal{E}_{\text{met}}$ and $\|g_\phi^{\text{de}} \circ g_\theta^{\text{en}}(s_t) - s_t\|$ in equation 10 together make $g_\theta^{\text{en}}$ a local isometric diffeomorphic representation. More specifically, for arbitrary points $s_1, s_2 \in M$, the metric is invariant under embedding $g_\theta^{\text{en}}$ such that $d_M(s_1, s_2) = d_N(g_\theta^{\text{en}}(s_1), (g_\theta^{\text{en}}(s_2))$, where $d_M$ and $d_N$ are the metric in the space $M$ and the latent space $N$ (Yano & Nagano, 1959). Isometry loss $\mathcal{E}_{\text{met}}$ is one of the key points to preserve the KEEC's control effect after mapping back to the original space. Without this loss function, the latent space may be distorted from its original, affecting the control consistency between the two spaces.

Finally, the loss is a linear combination of forward loss and isometry loss with a penalty $\lambda_{\text{met}} \in (0, 1)$:

$$\mathcal{E} = (1 - \lambda_{\text{met}})\mathcal{E}_{\text{fwd}} + \lambda_{\text{met}}\mathcal{E}_{\text{met}}. \tag{12}$$

$\mathcal{E}$ can be minimized by optimizing the parameters in the auto-encoder $g_\theta^{\text{en}}, g_\phi^{\text{de}}$ using stochastic gradient descent methods (See Algorithm 1 for details).

### 3.4 Optimal Control on Equivariant Vector Field

KEEC solves the control tasks on the latent space rather than in the original space. We follow model-based RL framework to conduct control (see Appendix C). Inspired by the Hamiltonian-Jacobi theory (Carinena et al., 2006), our control policy is based on a latent value function. Within this framework, the optimal control policy is determined by the vector field along the steepest ascent direction of the latent value function.

To perform control in the latent space, the value function should be invariant under the embedding $g$. This indicates that the control effect should be preserved under embedding. The lemma below proves this invariance.

**Lemma 3.3** (Invariant Value Function). *Under isometry embedding $g$, the value function is invariant to embedding $g$ for arbitrary policy $\pi$:*

$$V^\pi(s) = V^\pi(g(s)). \tag{13}$$

As shown by the research (Jean et al., 2019; Maslovskaya, 2018), the optimal control solutions remain consistent across both original and latent spaces under the isometry embedding $g$. Geometrically, the value function describes the integral of cumulative rewards along equivariant flow. When the embedding space is isometric to the original ones, it can directly lead to an invariant value function. With this invariance property, Lemma 3.3 demonstrates that we can solve the control problems based on the latent value function without mapping back to the original space. In the following, we denote the latent value function as $V_g^\pi := V^\pi \circ g$.

**Hamiltonian-Jacobi Optimal Value Function.** The value function and reward function defined on the latent space are represented as $V_g$ and $r_g$. With the *Bellman Optimality* $\mathcal{B}^*$, by using equation 4, $V_g$ can be

---

[2]Give two metric space $(X, d_X)$, $(Y, d_Y)$ and a function $g : X \to Y$. The distortion of $g$ is defined as $dis(g) = \sup_{x_1, x_2} \|d_X(x_1, x_2) - d_Y(g(x_1), g(x_2))\|$ (Federer, 2014). As an example, we see if $g$ is an isometry, the distortion is 0 so that $X$ and $Y$ are perfectly matched.

---

**Algorithm 1** KEEC: Learning

---

**Learning Equivariant Vector Field**

**Require:** Data $\mathcal{D} = \{\{\mathcal{T}_t^i\}_{t=t_0}^{t_0+L\Delta t}\}_{i=1}^{N_{\text{sample}}}$:
   transition tuple $\mathcal{T}_t^i = (s_t^i, a_t^i, r_t^i)$,
   time-interval $\Delta t$; learning rate $\alpha$;
   number of training epochs $K_{\text{flow}}$.
1: Initialize auto-encoder $g_{\theta^0}^{\text{en}}, g_{\phi^0}^{\text{de}}$
2: **for** Training epoch $k = 0, ..., K_{\text{flow}}$ **do**
3:    Map to the latent space $N$
      $\forall s \in \mathcal{D} : z = g_{\theta^k}^{\text{en}}(s)$
4:    Compute operators $\hat{\mathcal{P}}, \hat{\mathcal{U}}$ using equation 8.
5:    Compute the loss $\mathcal{E}$ by equation 12
6:    Update auto-encoder:
      $\theta^{k+1}, \phi^{k+1} = \theta^k + \alpha\nabla_\theta\mathcal{E}, \phi^k + \alpha\nabla_\phi\mathcal{E}$.
7: **end for**
8: **return** auto-encoder $\{g_\theta^{\text{en}}, g_\phi^{\text{de}}\}$
        operators $\{\hat{\mathcal{P}}, \hat{\mathcal{U}}\}$

**Learning Value Function**

**Require:** optimal state $s^*$; reward function $R$;
   number of training episodes $K_{\text{value}}$.
9: Initialize value net $\tilde{V}_g(\cdot, \psi^0)$
10: Initialize replay buffer $\mathcal{D}_{\text{Replay}} = \{\}$
11: **for** Training episodes $k = 0, ..., K_{\text{value}}$ **do**
12:    $z_0 = g_\theta^{\text{en}}(s_0)$
13:    **for** $t = 1, ..., T$ **do**
14:       Perform optimal policy equation 16
15:       Predict next state $z_{t+1}$ using equation 7
16:       Compute reward $r_t = R(g_{\phi^0}^{\text{de}}(z_t), a_t)$
17:       $\mathcal{D}_{\text{Replay}} = \mathcal{D}_{\text{Replay}} \cup (z_t, a_t, z_{t+1}, r_t)$
18:    **end for**
19:    Update value net with TD loss.
20: **end for**
21: **return** value net $\tilde{V}_g(\cdot, \psi)$

---

expressed as

$$\mathcal{B}^* V_g(z_t) = \max_{a_t \in \mathcal{A}} r_g(z_t, a_t) + \gamma V_g(\mathcal{K}_{\Delta t} z_t), \tag{14}$$

where $z_t = g(s_t)$. We apply temporal difference TD(0) to learn the latent value function parametrized by neural networks (more details in Appendix E). Then, we can obtain the analytical optimal policy from the learned value function. By the Hamiltonian-Jacobi theory (Carinena et al., 2006), the optimal action for $V_g$ in equation 14 can be transformed as

$$\max_{X_g} \mathcal{L}_{X_g} V_g(z_t), \tag{15}$$

where $X_g(z_t) := \mathcal{P} z_t + (\mathcal{U} z_t) a_t$ represents the corresponding latent vector field dependent on the action . When $X_g(z_t)$ points in the steepest ascent direction of $V_g(z_t)$, it will be the optimal control policy. In this scenario, we reframe the policy optimization problem as the optimization of the controlled vector field $X_g$. The derived analytical policy is in the following.

**Theorem 3.4** (Greedy Policy on Equivariant Vector Fields)**.** *Under Theorem 3.1 and Lemma 3.3, the optimal policy for the value function in the latent space has an analytical solution:*

$$\pi^*(z_t) = -[\nabla_a r_g(z_t, \cdot)]^\dagger (\gamma \nabla_z V_g^T \cdot \mathcal{U}(z))\Delta t \tag{16}$$

*where symbol $\dagger$ represents the inverse map with respect to a. (Proof in F.2; Corollary of quadratic-form latent value function in Appendix F.1).*

The analytical solution in equation 16 shows that the derived policy $\pi^*(z_t)$ is not only the optimal control solution in the latent space but also directly solves the maximization problem of the Hamilton–Jacobi theory (see equation 15). This highlights the necessity of our comprehensive embedding: the policy is obtained in analytical form via the learned invariant value function and equivariant vector fields, without the need to train a separate policy network. This theorem provides two main benefits. First, the analytical solution significantly reduces the time required to generate the optimal control policy (see the left side of Figure 3). Second, when this policy is incorporated into value iteration in equation 14, it ensures strong theoretical convergence to the optimal value function (see Appendix F.3). In practice, we can apply the automatic differentiation (Paszke et al., 2017) to compute the derivative $\nabla_z V_g$. We provided the pseudo algorithm for KEEC control in Algorithm 2 in Appendix G.

## 4 Numerical Experiments

In this section, we empirically evaluated the performances of KEEC in controlling the unknown nonlinear dynamical systems. We compared KEEC with four baselines: (a) Embed to control (E2C) - prediction, consistency, and curvature (PCC) (Levine et al., 2019); (b) Data-driven model predictive control (MPC) - model predictive path integral (MPPI) (Williams et al., 2017); (c) Online RL - soft actor-critic (SAC) (Haarnoja et al., 2018); (d) Offline RL - conservative Q-learning (CQL) (Kumar et al., 2020). These baselines covered a wide range of control methods for unknown dynamics, comprehensively investigating KEEC's control effectiveness. The experimental comparisons were conducted on a standard control benchmark - Gym and image-based pendulum- and two well-known physical systems - Lorenz-63 and wave equation.

**Control tasks.**  (1) **Pendulum** task involved swinging up and stabilizing a pendulum upright. We generated 1,000 trajectories; each has 50 steps with random controls using OpenAI Gym (Towers et al., 2023). In the **gym** version, the state was the pendulum's angle and angular velocity. In the **image** version, the same dynamics were simulated, but the state was defined as the image of the corresponding angle and angular velocity with $96 \times 48$ pixels (see Figure 6 in Appendix H.1). (2) **Lorenz-63 system**, a 3-dimension system known for its chaotic behaviour, was adapted with an affine controller acting on each dimension of the system. The goal was stabilizing the system on one of its strange attractors. In this environment, the state was defined by the system's three variables. We generated a dataset of 1,000 random control trajectories, each with 500 steps. (3) **Wave equation** is a second-order partial differential equation (PDE) system describing the wave propagation. The objective was stabilizing the waves to zero using ten controllers across the domain. The state was defined as the phase space, consisting of 50 states with their time derivatives. We generated 5,000 trajectories with random control using the controlgym (Zhang et al., 2023), each with 100 time steps (See more task details in Appendix H.1).

**Training Details.**  For each system, the reward $r$ was recorded as the quadratic reward[3] $r(s_t, a_t) = -(\|s_t - s^*\|_{R_2}^2 + \|a_t\|_{R_1}^2)$, where $s^*$ is the specified optimal state and $R_1, R_2$ are two positive definite matrices. The dataset was constructed by slicing the trajectories for training KEEC into multi-step $L = 8$, and slicing the trajectories for PCC, MPPI, and CQL into single-step $L = 1$, and then shuffling all slices. For the online algorithm SAC, the number of interactions with the environments is the same as the number of transition pairs in the offline data. All models were trained on the corresponding loss function using the Adam optimizer (Kingma & Ba, 2015). Appendix H.2 provides more details of the baselines and training details.

Table 1: Quantitative results. The results were the mean and standard deviation ($\pm$) of episodic rewards, evaluated with 100 initial states uniformly sampled from initial regions. We omitted the results of SAC, CQL, and MPPI on the image-based pendulum as their implementations did not support image inputs.

|  | Pendulum (OpenAI Gym | Image) | | Lorenz-63 | Wave Equation |
|---|---|---|---|---|
| SAC | $-95.1 \pm 48.7$ | N/A | $-4491.8 \pm 1372.4$ | $-1007.6 \pm 74.4$ |
| CQL | $-128.2 \pm 76.9$ | N/A | $-5782.5 \pm \mathbf{921.6}$ | $-4117.5 \pm 561.2$ |
| MPPI | $-187.2 \pm 78.7$ | N/A | $-8768.4 \pm 1831.1$ | $-34.5k \pm 2267.2$ |
| PCC | $-104.7 \pm 49.2$ | $-216.1 \pm 45.3$ | $-5123.6 \pm 1289.3$ | $-2249.2 \pm 133.6$ |
| KEEC (w/o $\mathcal{E}_{\text{met}}$) | $-852.3 \pm 128.7$ | $-205.7 \pm 33.7$ | $-8951.9 \pm 1927.4$ | $-28.9k \pm 3219.5$ |
| **KEEC** | $\mathbf{-94.9 \pm 44.8}$ | $\mathbf{-202.3 \pm 32.6}$ | $\mathbf{-2531.4 \pm} 1121.8$ | $\mathbf{-277.6 \pm 29.2}$ |

**Evaluation and Results.**  We reported the control performances in each system, particularly the mean and standard deviation of the episodic rewards. These results were evaluated with multiple runs using different random seeds and initial states sampled uniformly from specific regions. Episode lengths were set at 100 for the pendulum, 500 for Lorenz-63, and 200 for the wave equation. Additional details about the evaluation were detailed in Appendix H.3. Table 1 showed how KEEC outperformed the baseline algorithms by comparing the mean and standard deviations of the episodic rewards on the different control tasks. This phenomenon is

---

[3]The quadratic reward functions cover a broad range of RL problems. With the quadratic form, $R_1 \in \mathbb{R}^{m \times m}$ will be used in the numerical experiments.

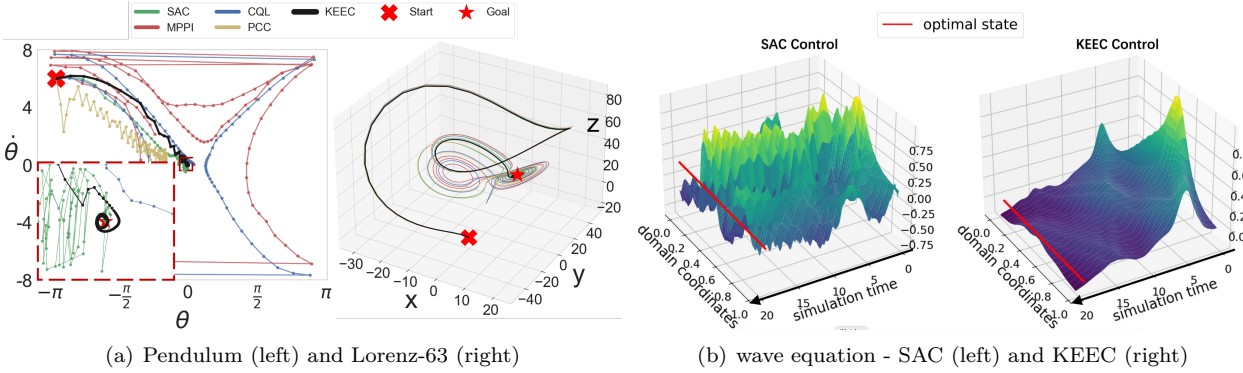

(a) Pendulum (left) and Lorenz-63 (right)          (b) wave equation - SAC (left) and KEEC (right)

Figure 2: Qualitative results. In (a-left), the Pendulum started at $(-3, 6)$ with a goal state of $(0, 0)$. A zoomed-in view of $[-0.1, 0.1] \times [-0.3, 0.3]$ showed control stability. In (a-right), the goal was the strange attractor $(-8, -8, 27)$ of Lorenz-63 system and we visualized the control trajectories of KEEC and the baselines. In (b), we showed the control trajectories of the KEEC in (b-right) and the best baseline SAC in (b-left). This task aimed to steer the system state to the zero state. Control trajectories of other baselines were shown in Appendix H.1.

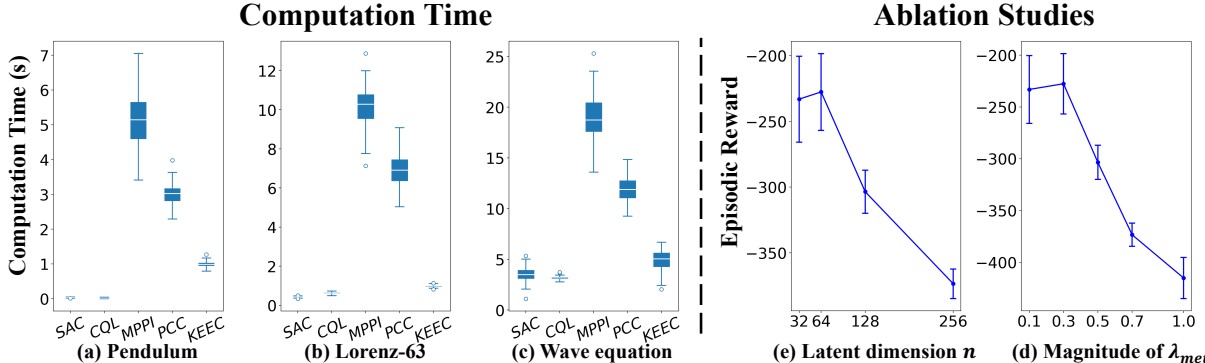

Figure 3: Quantitative results on evaluation time and ablation studies on latent dimension $n$ and magnitude of the isometric constraint $\lambda_{\mathrm{met}}$. Left: box-plots show the distributions of evaluation time. The white line in the box indicates the median. Our approach is consistently faster than the MPC-based methods and comparable to the RL methods. Right: different dimensions of the latent space (d) and our model's episodic reward with different magnitudes of $\lambda_{\mathrm{met}}$ (e).

more evident in Lorenz-63 and the wave equation; since their behaviours are highly nonlinear or even chaotic, MDP and simply locally linearized models can not sufficiently capture the pattern of dynamics.

In Figure 2, we presented the trajectories produced by various algorithms for three control tasks. By embedding vector field and metric information, KEEC improved control stability, as evidenced by the smooth trajectories and control robustness (Figure 2). Conversely, other baselines exhibited a "zig-zag" trajectory as they approached the goal state (see Figure 2(a)-left). In the Lorenz-63 task, the baselines' trajectories showed diverse control paths sensitive to minor perturbations due to the system's chaotic behaviour. KEEC, however, converged to the Lorenz attractor with minimal fluctuations (Figure 2(a)-right). This difference was because KEEC embedded vector field information (Figure 1(c)), enhancing control stability beyond control methods that rely on next-step predictions. For the wave equation task, we showed the control trajectories of best baseline-SAC and KEEC in Figure 2(b)). The results show that KEEC outperformed other algorithms, which struggled to effectively control a complex, nonlinear, and time-dependent field in high-dimensional PDE control. While SAC came closest to achieving success (see Figure 2(b)-left), it still failed to stabilize the phase space to zero. Figure 3(a-c) shows computation times for all methods. KEEC, with linear dynamics and an analytical control policy, is much faster than MPPI and PCC. Although computing the gradient of

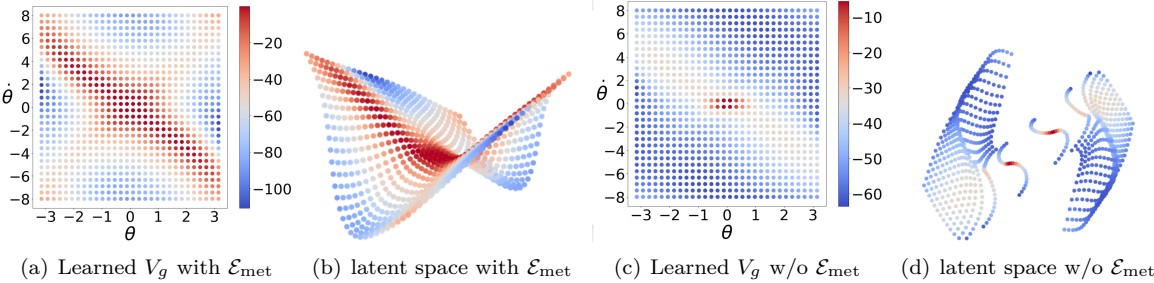

(a) Learned $V_g$ with $\mathcal{E}_{met}$    (b) latent space with $\mathcal{E}_{met}$    (c) Learned $V_g$ w/o $\mathcal{E}_{met}$    (d) latent space w/o $\mathcal{E}_{met}$

Figure 4: Comparison of learned value function $V_g$ and latent space with and without(w/o) the $\mathcal{E}_{met}$ in pendulum task. The colours on the original coordinates and the space indicated the magnitude of $V_g$. The spaces in (b) and (c) were visualized using Locally Linear Embedding (Roweis & Saul, 2000) to project from 8 to 3 dimensions.

the value net, $\nabla_z V_g$ via automatic differentiation makes KEEC slightly slower than MDP-based RL methods, the times remain comparable.

## 4.1 Ablation study

On top of comparing the performance of KEEC to the baselines, we revisited the wave equation control problem and performed an ablation analysis to assess the effects and sensitivity to (1) the magnitude of isometry loss $\lambda_{met}$ equation 11, and (2) the latent dimension $n$

**Latent dimension $n$.** In the main experiments, we set the latent dimension $n = 64$. To evaluate the model's performance under different latent dimensions, we varied the dimension from 32 to 256 while keeping other settings fixed. Figure 3(e) illustrates how the latent dimension $n$ affects the algorithm's performance. When $n$ is too small (e.g., 32), the latent space lacks sufficient capacity to fully capture the original dynamics linearly. Dimensions of 64 and 128 yield good control performance, but larger dimensions increase sample complexity, resulting in degraded performance with the same dataset.

**Isometry loss magnitude $\lambda_{met}$.** Figure 3(d) shows that our approach is robust for $\lambda_{met} \in [0.1, 0.3]$, though stronger constraints hinder learning control dynamics, indicating a trade-off. Control performance degraded significantly without constraints, as shown in Table 1. These results align with the theoretical analysis in Sections 3.2 and 3.3, emphasizing the need to preserve metrics for consistent control performance. Figure 4 visualizes the learned latent space for an inverted pendulum. With $\mathcal{E}_{met}$, the learned space was smooth by preserving the metric (Figure 4(b)), while without it, the space was distorted, and the optimal state cannot be observed (Figure 4(d)).

## 5 Conclusion

This paper introduces KEEC, a novel representation learning algorithm for unknown nonlinear dynamics control. By integrating principles from Lie theory and Koopman theory, KEEC constructs equivariant flows and vector fields. Because of the inherent equivariance and consistent metric, KEEC preserves the control effect across the original and latent space. Inspired by the Hamiltonian-Jacobi theory, KEEC utilizes the learned differential information to derive an analytical control policy, which improves computational efficiency and control robustness. We demonstrate these superiors in the numerical experiments, in which KEEC outperforms a wide range of competitive baselines.

**Limitations.** Our method relies on embedding the vector fields of the unknown dynamics to derive an analytical control policy to improve the control stability and avoid intensive numerical control optimization. Since the vector fields are characterized locally, we require the time step $\Delta t$ to be sufficiently small. Our approach may struggle with environments with a large time step $\Delta t$ (i.e., low observation frequency). An ablation study on how the magnitude of time step $\Delta t$ influences the control performance is required. In addition, our experiments on the image-based pendulum also demonstrated KEEC's effectiveness with image observations and potentially other types of observations. However, for handling different types of observations,

the design of the auto-encoder neural network is crucial in our approach. A generic auto-encoder design could degrade our method's performance in identifying and controlling dynamical systems.

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

## A Appendix

# A  Table of Notations

| Notations | Meaning |
|---|---|
| $a$ | action |
| $f$ | time derivative of the dynamical system |
| $g$ | isometry embedding |
| $g_\theta^{\mathrm{en}}$ | encoder |
| $g_\phi^{\mathrm{de}}$ | decoder |
| $\mathfrak{g}$ | Lie algebra |
| $s$ | state |
| $r$ | reward |
| $r_g$ | reward in latent space |
| $z$ | state in latent space |
| $\mathcal{A}$ | action space |
| $\mathcal{B}^*$ | Bellman optimality |
| $\mathcal{D}$ | Dataset |
| $F$ | flow of dynamical system |
| $M$ | original state space of dynamical system |
| $N$ | latent space |
| $GL$ | generalized linear group |
| $Hom(\cdot,\cdot)$ | homomorphic category |
| $\mathcal{K}$ | Koopman operator |
| $\mathcal{L}$ | Lie derivative |
| $\mathcal{P}$ | Lie algebra of one-parameter group |
| $r_1, r_2$ | reward functions related to action and state respectively |
| $\mathcal{S}$ | state space |
| $B$ | actuation matrix |
| $\mathcal{U}$ | actuation matrix |
| $V, V_g$ | value function and latent value function |
| $X, X_g$ | vector fields and equivariant (or latent) vector fields |
| $\Delta t$ | discretized time interval |
| $\lambda_{\mathrm{metric}}$ | penalty coefficient of isometric loss in loss function |
| $\lambda_{\mathrm{cost}}$ | penalty coefficient of action in reward function |
| $\theta$ | parameters of encoder |
| $\phi$ | parameters of decoder |
| $\psi$ | parameters of latent value function |
| $\pi$ | control policy |
| $\square^*, \square_*$ | pullback and pushforward symbols |

## B  Important Definitions

**Definition B.1** (Manifold, Riemannian Metric and Riemannian Manifold (Abraham et al., 2012)). A *manifold M* is a Hausdorff, second countable, locally Euclidean space. It is said of *dimension n* if it is locally Euclidean of dimension $n$. If a manifold with a globally defined differential structure, it is called a *smooth manifold* (Abraham et al., 2012). The manifold equipped a Riemannian product (Riemannian metric) structure $\langle \cdot, \cdot \rangle$ is called Riemannian manifold denoted by a pair $(M, \langle \cdot, \cdot \rangle)$.

**Definition B.2.** A *smooth function* of class $C^k$ is a function that has continuous derivatives up to the $k$-th order. Specifically, if a function $f : U \to \mathbb{R}$ (where $U$ is an open subset of $\mathbb{R}^n$) is said to be of class $C^k$, it satisfies the following properties:

- The function $f$ has **partial derivatives** of all orders from 1 up to $k$.

- These partial derivatives are **continuous** up to the $k$-th order.

In formal terms, a function $f$ is of class $C^k$ if:

$$f \in C^k(U) \quad \text{if} \quad \frac{\partial^{|\alpha|} f}{\partial x_1^{\alpha_1} \cdots \partial x_n^{\alpha_n}} \quad \text{exists and is continuous for all multi-indices} \quad |\alpha| \le k,$$

where $\alpha = (\alpha_1, \ldots, \alpha_n)$ is a multi-index representing the orders of partial differentiation.

**Definition B.3** (Group (Druţu & Kapovich, 2018)). A group is non-empty set $G$ with a binary operation on $G$, here denoted as "$\cdot$", then the group can be written as $(G, \cdot)$, three axioms need to be satisfied on group:

- *Associativity.* for all $a, b, c$ in $G$, one has $(a \cdot b) \cdot c = a \cdot (b \cdot c)$;

- *Identity Element.* There exists an element $e$ in $G$ such that, for every $a$ in $G$, one has $e \cdot a = a$ and $a \cdot e = a$, such an element is unique in a group;

- *Inverse Element.* For each element $a$ in $G$, there exists an element $b$ in $G$ such that $a \cdot b = e$, the $b$ is unique commonly denoted as $a^{-1}$.

**Definition B.4** (Vector Field). Let $U \subseteq \mathbb{R}^n$ be an open subset. A *vector field* on $U$ is a smooth (continuously differentiable) function

$$f : U \to \mathbb{R}^n$$

that assigns to each point $s = (s_1, s_2, \ldots, s_n) \in U$ a vector

$$f(s) = (f_1(s), f_2(s), \ldots, f_n(s)).$$

**Definition B.5** (Lie Group Action). Let $G$ be a Lie group and $M$ be a smooth manifold. A Lie group action of $G$ on $M$ is a group action:

$$\sigma : G \times M \to M, \quad (g, s) \mapsto g \cdot s \tag{17}$$

such that the action map $\sigma$ is smooth.

**Definition B.6** (Flow). Given a vector field $f$ on an open subset $U \subseteq \mathbb{R}^n$, a *flow* generated by vector field $f$ is a family of diffeomorphisms

$$F_t : U \to U, \quad t \in \mathbb{R},$$

satisfying the following properties:

1. **Initial Condition**: For all $s \in U$,
$$F_0(s) = s.$$

2. **Group Property**: For all $s, t \in \mathbb{R}$ and $\mathbf{x} \in U$,
$$F_{s+t}(s) = F_s(F_t(s)).$$

3. **Differential Equation**: For each $s \in U$, the curve $\phi(t) = F_t(s)$ is a solution to the ordinary differential equation
$$\frac{d}{dt}\phi(t) = f(\phi(t)),$$
with initial condition $\phi(0) = s$.

Typically, the flow map $F$ can be treated as a local one-parameter Lie group of parameterized by $t$.

**Definition B.7** (Lie Derivative (Lie, 1893))**.** The Lie derivative of a function $g : M \to \mathbb{R}$ with respect to vector field $X$ at a point $s \in M$ is the function

$$(\mathcal{L}_X g)(s) = \lim_{t \to 0} \frac{g(\phi_s(t)) - g(s)}{t}, \tag{18}$$

where $\phi_s(t)$ the flow through $s$.

**Definition B.8** (Infinitesimal Generator)**.** Suppose $\phi : \mathbb{R} \to GL(n, \mathbb{R})$ is a one-parameter group. Then there exists a unique $n \times n$ matrix $X$ such that $\phi(t) = \exp(tX)$ for all $t \in \mathbb{R}$. It follows from the result that is differentiable, and the matrix $X$ can then be recovered from $\phi$ as

$$\left.\frac{d\phi(t)}{dt}\right|_{t=0} = \left.\frac{d}{dt}\right|_{t=0} \exp(tX) = \left.(X \exp(tX))\right|_{t=0} = X \exp(0) = X, \tag{19}$$

where $X$ is called *infinitesimal generator*.

## C  Model-based RL

The most well-known and popular reinforcement learning methods are model-free (Lillicrap, 2015; Mnih et al., 2015; Schulman, 2015; Schulman et al., 2015; 2017; Vinyals et al., 2019; Ma et al., 2021). Model-free RL has been widely used in many areas and manage to optimize the performance of models especially in language tasks (Achiam et al., 2023; Touvron et al., 2023; Ouyang et al., 2022). Despite of the generalization ability and scalability of model-free RL, when it comes to applying to continuous control, results from the traditional control theory and the induced analytical closed-form policy extraction, which can potentially be used to reduce the sample complexity, have been ignored. In this work, we leverage these missing advantages to develop our algorithm under the model-based RL framework. In addition, model-based RL is more suitable to be used in the setting of offline learning. Model-free offline RL is well-explored (Kumar et al., 2020; Kostrikov et al., 2021; Fujimoto & Gu, 2021; Chen et al., 2021; Tang et al., 2024). However, model-based offline RL has an advantage that the offline data can be used to learn the dynamics once, and transferred to other tasks. For instance, in the tasks of swing-up and balance in inverted-pendulum (Todorov et al., 2012), where these two tasks have different starting points, but share the same dynamics.

Based on the dynamics equation 1, a model-based RL decision-making process is provided. The sequential actions $\{a_\tau\}$ are determined by a policy $\pi(\cdot \mid s_\tau)$, the target is to maximize the expected accumulated reward $r : M \to \mathbb{R}$ in the future, such that

$$V^\pi(s_t) = \mathbb{E}[\sum_\tau \gamma^\tau r(s_\tau, a_\tau) \mid s_\tau, a_\tau \sim \pi], \quad \forall s_t \in M, \tag{20}$$

where $V^\pi : M \to \mathbb{R}$ is the value function to measure the future expected accumulated reward for the arbitrary state, $\gamma \in (0,1)$ is the discounted factor, and $s_0 \in M$ is the initial state.

The control part of KEEC is derived from model-based RL (Okada & Taniguchi, 2020; Mittal et al., 2020). The algorithm aims to search an action sequence in an infinite horizon as a control policy, expressed as $a_{t:t+(L-1)\Delta t} = (a_t, a_{t+\Delta t}, \cdots, a_{t+(L-1)\Delta t})$. The optimal actions under model-based RL can be defined as:

$$\underbrace{a^*_{t:t+(L-1)\Delta t}}_{\text{search optimal policy for } L-\text{step rollout}} \in \arg\max \quad \mathbb{E}[\underbrace{\sum_{\tau=t}^{t+(L-1)\Delta t} \gamma^\tau r(s_\tau, a_\tau)}_{L-\text{step rollout}} + \gamma^L V(s_{t+L\Delta t})], \tag{21}$$

$$s.t. \quad s_{\tau+\Delta t} = F_{\Delta t}(s_\tau, a_\tau), \quad \forall \tau \in \{t, t+\Delta t, \cdots, t+(L-1)\Delta t\}, \ \forall \ a_\tau \in \mathcal{A}.$$

### C.1  Bellman Optimality

The value function approximation in the model-based RL framework is considered as a control certificate. Based on the original work (Bellman, 1966), the $N-$step look-forward latent value function can be represented as

$$\mathcal{B}^* \tilde{V}_g^k(z_t) = \max_{a_{t:t+(L-1)\Delta t}} \sum_{\tau=t}^{t+(L-1)\Delta t} \gamma^\tau r_g(z_\tau, a_\tau) + \gamma^L \tilde{V}_g^k(z_{t+L\Delta t}), \quad \forall s_t \in S, \tag{22}$$

where $B^*$ is the Bellman optimality and $k+1, k \in \mathbb{L}^+$ are the number of iterations of the approximated value function $\tilde{V}_g$. It is known that the approximated value function contracts to a fixed point (Bertsekas, 2012; Lagoudakis & Parr, 2003), such that

$$\limsup_{k\to\infty} \|\tilde{V}_g^k - V_g^*\|_\infty \le \epsilon,$$

where $V_g^*$ is the optimal value function and $\epsilon$ is arbitrary small.

In original dynamic programming methods, the search for optimal actions relies on the discretization of the state space. However, the number of states and actions increases exponentially with refinement, and optimization eventually becomes a costly computational problem. To overcome the drawbacks of other embedding control works, the analytical form of the optimal policy can be derived by discovering the vector fields on the latent space.

# D  Koopman Semigroup and Equivariance

**Koopman Semigroup (Das, 2023).**  The Koopman operator $\mathcal{K}_t$ acts on a function space by composition with the flow map $F_t$, effectively implementing time shifts in function $g$. Various choices exist for the function space, such as $L^2(M)$ and spaces of continuous functions. In this context, we restrict our attention to $C^1(M)$, the space of once continuously differentiable functions on the space $M$. Specifically, for a function $g \in C^1(M)$ and time $t \in \mathbb{R}$, the Koopman operator $\mathcal{K}_t : C^1(M) \to C^1(M)$ is defined as:

$$(\mathcal{K}_t g)(s) = g(F_t(s)),$$

where $F_t : M \to M$ is the flow map with interval time $t$.

In general, if $F_t$ is a $C^k$ flow for some $k \geq 1$, then $\mathcal{K}_t$ maps the space $C^r(M)$ into itself for every $0 \leq r \leq k$. The infinitesimal generator $\mathcal{P}$ of the Koopman operator $\mathcal{K}_t$ is defined by:

$$\mathcal{P}g := \lim_{t \to 0} \frac{1}{t}(\mathcal{K}_t g - g), \quad g \in \mathcal{D}(\mathcal{P}),$$

where $\mathcal{D}(\mathcal{P}) \subseteq C^1(M)$ is the domain of $\mathcal{P}$, consisting of functions for which this limit exists. Typically, $\mathcal{P} : C^1(M) \to C^0(M)$ when $F_t$ is sufficiently smooth.

Notably, $\mathcal{P}$ is an unbounded operator on $C^1(M)$, meaning it is not defined on the entire space but rather on a dense subset $\mathcal{D}(\mathcal{P})$. This dense subset ensures that $\mathcal{P}$ can approximate its behavior across $C^1(M)$ through limits of convergent sequences within $\mathcal{D}(\mathcal{P})$.

Furthermore, when considering $C^1(M) \cap L^2(M)$, the action of the extended generator $\hat{\mathcal{P}}$ coincides with that of $\mathcal{P}$ on this intersection, ensuring consistency across different function spaces. According to the strong operator topology, the Koopman semigroup $\{\mathcal{K}_t\}$ can be approximated by $\exp(t\mathcal{P}_n)$ for a Cauchy sequence $\{\mathcal{P}_n\}$ of bounded operators converging to $\mathcal{P}$ on $\mathcal{D}(\mathcal{P})$. Sepcifically, for each $g \in C^1(M) \cap L^2(M)$, we have $\lim_{n \to \infty} \|\mathcal{K}g - \exp(\mathcal{P}_n t)g\|_2 \to 0$.

The family of Koopman operators $\{\mathcal{K}_t\}_{t \in \mathbb{R}}$ forms a one-parameter group of linear operators, satisfying:

$$\mathcal{K}_{t+s} = \mathcal{K}_t \circ \mathcal{K}_s, \quad \mathcal{K}_0 = \text{Identity operator.}$$

This group structure implies that $\mathcal{K}_t = \exp(t\mathcal{P})$, where $\exp(t\mathcal{P})$ is defined via the operator exponential for the generator $\mathcal{P}$.

The generator $\mathcal{P}$ acts as a differentiation operator:

$$\mathcal{P}g = \mathcal{L}g,$$

where $X$ is the vector field defining the flow $F_t$, and $\mathcal{L}$ denotes the Lie derivative of $g$ along $X$. Consequently, $\mathcal{P}$ generates the vector fields of $g$ in the function space $C^1(M)$, facilitating a linear representation of the potentially nonlinear system.

**Group Representation and Equivariance.**  Let $(\rho, V)$ and $(\tau, W)$ be group representations of $C$. A linear map $g : V \to W$ is called $C-$linear map if $g(\rho(c)v) = \tau(c)g(v)$ for any $v \in V$ and $c \in C$, that is if the diagram in D commutes (Koyama et al., 2023).

$$
\begin{array}{ccc}
V & \xrightarrow{\rho(c)} & V \\
\downarrow{g} & & \downarrow{g} \\
W & \xrightarrow{\tau(c)} & W
\end{array}
$$

A $C-$linear map is a homomorphism of the representation of $C$. If there is a bijective $C-$map between two representations of $C$, they $C-$isomorphic, or isomorphic for short. When the isomorphism exists, the two representations $\rho$ and $\tau$ are said to be completely equivalent.

**Discussion of Equivariance of Flow and Vector Fields under Koopman Operator.** Back to our case, our target is to construct the representation of one-parameter group - flow map. The flow $F_t : M \to M$ generated by the vector fields is a diffeomorphism $\mathrm{Diff}(M)$ that can be equivalently described in the Koopman framework by the automorphism $\mathcal{K}_t$. According to infinite-dimensional Lie group theory, when $M \subset \mathbb{R}^m$ is compact, the group of diffeomorphisms $\mathrm{Diff}(M)$ can be endowed with a Fréchet-Lie group structure. Similarly, the group of automorphisms $Aut(C^\infty(M))$ [4] also forms a Fréchet-Lie group. Under the topology of local uniform convergence of all partial derivatives on $C^\infty(M)$, there exists an isomorphism between these two Fréchet-Lie groups. This isomorphism preserves both the group structure and the smooth manifold structure, establishing a deep connection between diffeomorphisms of $M$ and automorphisms of the function space. The associated Lie algebras of these groups are the vector fields $Vect(M)$ for $\mathrm{Diff}(M)$ and the derivations (generator) $Der(C^\infty(M))$ [5] [6] for $Aut(C^\infty(M))$. The isomorphism between $\mathrm{Diff}(M)$ and $Aut(C^\infty(M))$ induces an isomorphism between their respective Lie algebras, $Vect(M)$ and $Der(C^\infty(M))$. This correspondence implies that a vector field $f \in Vect(M)$ on the manifold $M$ is equivalent to a derivation $\mathcal{P} \in Der(C^\infty(M))$ on the function space, defined by $\mathcal{P}g = \nabla g \cdot f$ for all $g \in C^\infty(M)$. The detailed proof can be found in (Omori, 2017; Schmid, 2004). Consequently, the flow maps $F_t$ and the Koopman operators $\mathcal{K}_t$ are intertwined by this isomorphism, making them equivariant under the constructed representations. Similarly, the vector fields $f \in Vect(M)$ and the derivations $\mathcal{P} \in Der(C^\infty(M))$ are equivariant under the induced representations. Please be note that the local flow $F_t$ can always be mollified to a smooth function with arbitrary small error (Pedregal, 2024), and thus above isomorphic map $g$ can always be found.

**Convergence Property.** Our method distinguishes itself from previous work by focusing on the learning of the infinitesimal generator $\mathcal{P}$. One advantage of this approach is the ability to approximate and capture mixed spectra with strong convergence. We consider the Koopman operator $\mathcal{K}_t$, which acts on the square-integrable space $L^2(M) = H_c \oplus H_p$, where $H_c$ and $H_p$ represent the continuous and atomic (point) spectra, respectively. The generator $\mathcal{P}$ of the Koopman operator is a densely defined, unbounded operator with domain $\mathcal{D}(\mathcal{P}) \subset L^2(M)$. In our approach, we approximate $\mathcal{P}$ by constructing a compactified version, $\hat{\mathcal{P}}$, following the compactification procedure described in (Das et al., 2021). Specifically, it is shown in (Das et al., 2021) that the operator $\hat{\mathcal{P}} = \Pi\mathcal{P}\Pi$ is a compact operator with a purely atomic spectrum, providing an approximation to the original unbounded generator $\mathcal{P}$. Here, $\Pi$ is a projection operator that maps $L^2(M)$ to the feature function space spanned by $g$, which is dense and countable in $L^2(M)$. The approximated operator $\hat{\mathcal{P}}$ can be expressed as $\hat{\mathcal{P}} = \lim_{t \to 0^+} \frac{\Pi\mathcal{K}_t\Pi - I}{t}$, consistent with our learning process as described in Equations (8) and (9) of our work. Moreover, $\hat{\mathcal{P}}$ achieves strong convergence in operator topology to $\mathcal{P}$ as $t \to 0^+$, implying that the spectral properties of $\hat{\mathcal{P}}$ approximate those of $\mathcal{P}$. This convergence also ensures that the spectral measures of $\hat{\mathcal{P}}$ approximate those of $\mathcal{P}$, effectively capturing both the atomic and continuous components of the Koopman spectrum. Consequently, the approximated Koopman evolution operator $\exp\left(\hat{\mathcal{P}}t\right)$ achieves strong convergence to $\mathcal{K}_t$, even when the Koopman operator has a mixed spectrum. This result is supported rigorously by Corollary 4 in (Das et al., 2021), highlighting the quality of the approximation.

## E  Additional Details of Value Learning

The parameter $\psi$ is optimized using stochastic gradient descent to minimize the following loss for value learning:

$$\mathcal{E}_{\mathrm{evf}} = \sum_{(z_t, a_t^*, z_{t+\Delta t}) \in \mathcal{D}_{\mathrm{Replay}}} \|r_g(z_t, a_t^*) + \gamma\tilde{V}_g(z_{t+\Delta t}, \psi) - \tilde{V}_g(z_t, \psi)\|, \tag{23}$$

where $a^* = \pi^*(z_t)$ and $\mathcal{D}_{\mathrm{Replay}}$ is the replay buffer storing the system trajectories under the control policy $\pi^*$ in equation 16. The analytical form of the optimal action is performed directly on the continuous space

---

[4]$Aut(C^\infty(M))$ refers to the set of all automorphisms (structure-preserving bijective maps) of the space of smooth functions $C^\infty(M)$ on space $M$. This group consists of all maps that preserve the algebraic and differentiable structure of the space of smooth functions.

[5]The Lie algebra of derivations $Der(C^\infty(M))$ associated with the group of automorphism $Aut(C^\infty(M))$ is a mathematical structure that describes the infinitesimal transformations of the space of smooth functions $C^\infty(M)$ on manifold $M$.

[6]A derivation on $C^\infty(M)$ is a linear map $D : C^\infty(M) \to C^\infty(M)$ that satisfies the Leibniz rule: $D(fg) = fD(g) + gD(f) \quad \forall f, g \in \mathcal{C}^\infty(M)$.

via learned equivariant vector fields equation 6. In addition to computational efficiency, the convergence of $\tilde{V}_g(\cdot, \psi)$ enables a fast convergence rate to $V_g^{\pi^*}$, as shown in the following theorem.

**Theorem E.1** (Quadratic Convergence of Value Functions). *When Theorem 3.4 holds, the approximated latent value function $\tilde{V}_g(\cdot, \psi)$ will point-wisely converge to the optimal $V_g^{\pi^*}$, and the convergence rate is quadratic as follows:*

$$\|\tilde{V}_g^{k+1} - V_g^{\pi^*}\| = \mathcal{O}(\|\tilde{V}_g^k - V_g^{\pi^*}\|^2), \tag{24}$$

*where the $k_{th}$ update of $\tilde{V}_g^k$ is calculated by minimizing $\mathcal{E}_{evf}$ in equation 23.*

## F   Proofs of Main Theorems

This section provides proof of the lemma and theorems in the main text.

### F.1   Proof of Theorem 3.1.

*Proof.* Before proving the lemma, the property of pushforward of diffeomorphism needs to be given.

**Pushforward map (Ma & Fu, 2012).** Let $g$ be an embedding map $g : M \to N$, for a vector field $X \in Vect(M)$ and flow $\phi : I \to M$ and corresponding vector field $(g_* X)(\phi(t)) \in Vect(N)$, there exists

$$(g_* X)\big(\phi(t)\big) = d\Big(g\big(\phi(t)\big)\Big)X(t), \tag{25}$$

where $g_*$ is the pushforward operator.

Therefore, the pushforward can be represented as

$$\begin{aligned}
\frac{d}{dt}(\mathcal{K}_t^{a_t} g)(s_t) &= d(g(s_t))f(s_t, a_t) \\
&= \nabla(g(s_t)) \cdot [f_M(s_t) + B(s_t)a_t] \\
&= \frac{\partial g}{\partial s}(s_t)f_M(s_t) + \frac{\partial g}{\partial s}(s_t)B(s_t)a_t \\
&= \underbrace{\mathcal{P}z_t}_{\text{embedding of drift part}} + \underbrace{\mathcal{U}(z_t)a_t}_{\text{embedding of actuation part}},
\end{aligned} \tag{26}$$

where $\mathcal{P}z_t$ is the equivariant vector field without any action intervention. The $\frac{\partial g}{\partial s}(s_t)B(s_t)$ is the embedding of the actuation matrix. Under the certain embedding condition, we can simplify the acutation part to a interaction form as $(\mathcal{U}z_t)a_t$, where $\mathcal{U}$ is a three-mode tensor. $\square$

According to the Theorem 3.1, the flow can be represented by the exponential map in Lie theory

$$\begin{aligned}
z_{\Delta t} &= \exp(\mathcal{P}\Delta t)\big(z_0 + \int_0^{\Delta t} \exp(-t\mathcal{P})\mathcal{U}(z_t)a_t dt\big) \\
&= \exp(\mathcal{P}\Delta t)z_0 + \int_0^{\Delta t} \exp(\mathcal{P}[\Delta t - t])\mathcal{U}(z_t)a_t dt \\
&\approx \exp(\mathcal{P}\Delta t)z_0 + \mathcal{P}^{-1}(\exp(\mathcal{P}\Delta t) - I)\mathcal{U}(z_0)a_0
\end{aligned} \tag{27}$$

where the first line is the solution of non-homogeneous linear ODE (see Equation 2.61 on page 20 (Schekochihin, 2022)). When the $\Delta t$ is sufficiently small, the right-side of the third line can be approximated as $\mathcal{P}^{-1}(\exp(\mathcal{P}\Delta t) - I)\mathcal{U}(z_t)a_t$. Then the difference between $z_{\Delta t}$ and $z_0$ can be

$$\begin{aligned}
&z_{\Delta t} - z_0 \\
&= \exp(\Delta t\mathcal{P})z_0 + \mathcal{P}^{-1}(\exp(\mathcal{P}\Delta t) - I)\mathcal{U}(z_0)a_0 - z_0 \\
&= (I + \mathcal{P}\Delta t + \sum_{n \geq 2} \frac{1}{n!}\frac{d^n(\mathcal{K}_{\Delta t})}{dt^n}\Delta t^n - I)z_0 + \mathcal{P}^{-1}(I + \mathcal{P}\Delta t + \sum_{n \geq 2} \frac{1}{n!}\frac{d^n(\mathcal{K}_{\Delta t})}{dt^n}\Delta t^n - I)\mathcal{U}(z_0)a_0 \\
&= \mathcal{P}z_0\Delta t + \mathcal{U}(z_0)a_0\Delta t + \mathcal{O}(\Delta t^2).
\end{aligned} \tag{28}$$

### F.2   Proof of Theorem 3.4 and Corollary F.1.

The proof is directly developed from the dynamic programming (Bertsekas, 2012) and the Hamilton-Jacobi-Bellman (HJB) equation (Yong & Zhou, 1999).

In this case, the proof will be decomposed into two cases, one to prove a general case and another to prove the solution of the special quadratic form of the reward function (covering a wide range of Linear Quadratic Regulator problems).

**1. Proof of Theorem 3.4**   By the Bellman optimality, the optimal value function can be represented as a similar form in Equation equation 14:

$$B^*V(s_t) = \max_{a_t} r(s_t, a_t) + \gamma V(F_{\Delta t}(s_t)). \tag{29}$$

According to the Lemma 3.1 and Theorem 3.3, there exists a representation in latent space as

$$B^*V_g(z_t) = \max_{a_t} r_g(z_t, a_t) + \gamma V_g(\mathcal{K}_{\Delta t}(z_t)). \tag{30}$$

In this scenario, it assumes that the value function $V \in C^1(M, \mathbb{R})$, then $V_g \in C^1(N, \mathbb{R})$.

By the definition of the HJB equation (Yong & Zhou, 1999), the standard form exists:

$$V(x(t + \Delta t), t + \Delta t) = V(x(t), t) + \frac{\partial V(x(t), t)}{\partial t}\Delta t + \frac{\partial V(x(t), t)}{\partial x} \cdot \dot{x}(t)\Delta t + o(\Delta t), \tag{31}$$

where $x(t)$ is the state at time $t$, since in the value function is time-independent, the $\frac{\partial V(x(t),t)}{\partial t} = 0$. Back to the case, the integral form can be obtained as:

$$V_g(\mathcal{K}_{\Delta t}(z_t)) = V_g(z_t) + \int_0^{\Delta t} \mathcal{L}_{X_g} V_g(z_\tau) d\tau + o(\Delta t)$$
$$\approx V_g(z_t) + \nabla_{z_t} V_g^T(z_t) \cdot X_g(\phi(t))\Delta t + o(\Delta t) \tag{32}$$

The Lie derivative $\mathcal{L}_{X_g} V_g(z_\tau)$ interprets the change value function $V_g(z_\tau)$ under the vector field of $X_g(z_t) \in Vect(N)$. Instead of searching for a direction in Euclidean space, $\mathcal{L}_{X_g} V_g(z_\tau)$ can be understood as the change of value function along the tangent vector $X_g(z_t)$ (Wu & Duan, 2021) on the latent space. When the $\Delta t$ is sufficiently small, the second line of equation 32 holds. Observing the right-hand side of equation 30 can be replaced by Equation equation 32, the following equation can be obtained as

$$B^*V_g(z_t) = \max_{a_t} r_g(z_t, a_t) + \gamma V_g(\mathcal{K}_{\Delta t}(z_t))$$
$$= \max_{a_t} r_g(z_t, a_t) + \gamma(V_g(z_t) + \nabla_{z_t} V_g^T(z_t) \cdot X_g(z_t)\Delta t + o(\Delta t)) \tag{33}$$
$$= \max_{a_t} r_g(z_t, a_t) + \gamma V_g(z_t) + \gamma \nabla_{z_t} V_g^T(z_t)[\mathcal{P}z_t + \mathcal{U}(z_t)a_t]\Delta t + o(\Delta t),$$

where the vector field of $X_g(\phi(t))$ is defined as in equation 6. Typically, the reward function can be decomposed as two separable functions defined on state and action, respectively. Here, the $r_g$ can be defined as

$$r_g(z, a) = r_1(a) + r_2(z), \tag{34}$$

where $r_1$ and $r_2$ are two independent functions. Plug-in the equation 32 into the equation 33, the following form exists:

$$\max_{a_t} r_g(z_t, a_t) + \gamma\Big(V_g(z_t) + \nabla_{z_t} V_g^T(z_t)[\mathcal{P}z_t + \mathcal{U}(z_t)a_t]\Delta t + o(\Delta t)\Big)$$
$$= \max_{a_t} r_g(z_t, a_t) + \gamma V_g(z_t) + \gamma \nabla_{z_t} V_g^T(z_t)[\mathcal{P}z_t + \mathcal{U}(z_t)a_t]\Delta t + o(\Delta t) \tag{35}$$
$$= \max_{a_t} \underbrace{r_g(z_t, a_t) + \gamma \nabla_{z_t} V_g^T(z_t)[\mathcal{P}z_t + \mathcal{U}(z_t)a_t]\Delta t}_{\text{depedent on action}} + \gamma V_g(z_t) + o(\Delta t).$$

When the $r_g$ is a convex function, the optimization becomes a convex problem, which can be solved analytically.

Since $r_g$ is quadratic form, $\nabla_a r_g(z_t, \cdot)$ is a linear operator, and thus $\nabla_a r_g(z_t, a) = \nabla_a r_g(z_t, \cdot)a$. The equation 35 can be solved by taking the gradient equal zero:

$$\max_{a_t} r_g(z_t, a) + \gamma \nabla_{z_t} V_g^T(z_t)[\mathcal{P}z_t + \mathcal{U}(z_t)a_t]\Delta t$$
$$\Rightarrow \quad \nabla_a r_g(z_t, a) + \gamma \nabla_{z_t} V_g^T(z_t)\mathcal{U}(z_t)\Delta t = 0 \tag{36}$$
$$\Rightarrow \quad a_t^* = -[\nabla_a r_g(z_t, \cdot)]^\dagger(\gamma \nabla_{z_t} V_g^T \cdot \mathcal{U}(z_t))\Delta t,$$

**Corollary F.1** (Analytical form of value function with quadratic form). *Under Theorem 3.4, if the value function $V_g$ has a quadratic form, i.e., $V_g(z) = -(z - z^*)^T W(z)(z - z^*) + b$ with $W(z)$ being symmetric positive definite matrix, $b$ being a constant, then, the optimal action is:*

$$\pi^*(z_t) = -\gamma R_1^\dagger [z_t^T W(z_t) + \frac{1}{2}\gamma(z_t - z^*)^T \frac{\partial W(z_t)}{\partial z}(z_t - z^*)]\mathcal{U}(z_t). \tag{37}$$

*Proof.* For the reward function, the equation 34 becomes as

$$r_g(z, a) = -a^T R_1 a \Delta t - z^T R_2 z \Delta t, \tag{38}$$

where $R_1$ and $R_2$ are symmetric semi-positive definite matrices, and their dimensions rely on the dimension of observables and actions. In this case, the value function will also become a quadratic form as

$$V(z) = -(z - z^*)^T W(z)(z - z^*) + b, \tag{39}$$

where $z^*$ is the target state in observable space and $M(z)$ is a positive definite matrix. Correspondingly, the equation 32 in quadratic form becomes

$$
\begin{aligned}
V_g(\mathcal{K}_t(z_t)) &= V_g(z_t) + \nabla_{z_t} V_g^T(z_t) \cdot X_g(z_t)\Delta t + o(\Delta t) \\
&= -(z_t - z^*)^T W(z_t)(z_t - z^*) - 2z_t^T W(z_t) \cdot [\mathcal{P}z_t + \mathcal{U}(z_t)a_t]\Delta t \\
&\quad - (z_t - z^*)^T \frac{\partial W(z_t)}{\partial z}(z_t - z^*) \cdot [\mathcal{P}z_t + \mathcal{U}(z_t)a_t]\Delta t + o(\Delta t).
\end{aligned} \tag{40}
$$

Meanwhile, the equation 33 can be

$$
\begin{aligned}
B^* V_g(z_t) &= \max_{a_t} r_g(z_t, a_t) + \gamma V_g(z_t) + \gamma \nabla_{z_t} V_g^T(z_{t+1})[\mathcal{P}z_t + \mathcal{U}(z_t)a_t]\Delta t + o(\Delta t) \\
&= \max_{a_t} -a_t^T R_1 a_t \Delta t - z_t^T R_2 z_t \Delta t - \gamma(z_t - z^*)^T W(z_t)(z_t - z^*) - 2\gamma z_t^T W(z_t) \cdot [\mathcal{P}z_t + \mathcal{U}(z_t)a_t]\Delta t \\
&\quad - \gamma(z_t - z^*)^T \frac{\partial W(z_t)}{\partial z}(z_t - z^*) \cdot [\mathcal{P}z_t + \mathcal{U}(z_t)a_t]\Delta t + o(\Delta t).
\end{aligned}
$$
$$\tag{41}$$

Due to the convexity of the equation 41, taking the gradient equal to zero can yield the optimal action as

$$
\begin{aligned}
&- 2R_1 a_t - 2\gamma z_t^T W(z_t)\mathcal{U}(z_t) - \gamma(z_t - z^*)^T \frac{\partial W(z_t)}{\partial z}(z_t - z^*)\mathcal{U}(z_t) = 0 \\
\Rightarrow \quad &a_t^* = -\gamma R_1^\dagger [z_t^T W(z_t) + \frac{1}{2}(z_t - z^*)^T \frac{\partial W(z_t)}{\partial z}(z_t - z^*)]\mathcal{U}(z_t).
\end{aligned} \tag{42}
$$

### F.3 Proof of Invariant Value Learning Convergence

Approximation in value space with one-step greedy TD amounts to a step of Newton's method for solving HJB equation. In the following proof, the convergence of the latent value function $\{\tilde{V}_g(\psi^k)\}_{k=1}^\infty$ is treated as a Cauchy net in the functional space contracting to the fixed point $V_g^{\pi^*}$. The proof has a natural connection to Newton-Raphson method (Wright, 2006).

**Lemma F.2** (Newton-Raphsom Method (Nocedal & Wright, 1999)). *Consider a contraction map $E : Y \to Y$ and the fixed point $y^* = \lim_{n \to \infty} E^n(y^0)$ for some initial vector $y^0 \in Y \subset \mathbb{R}^n$ and $y^* = E(y^*)$. The step-wise difference is defined as*

$$D(y^k) = E(y^k) - y^k \tag{43}$$

*where $\forall y^k \in \mathcal{C}^2$ and the contraction operator $E$ indicates the fact that $\lim_{k \to \infty} D(y^k) \to 0$, the Newton's step is to update $y^{k+1}$ as*

$$y^{k+1} = y^k - [\nabla D(y^k)^T]^{-1} D(y^k) \tag{44}$$

*where $D(y^k)$ is differentiable and the $\nabla D(y^k)$ is an invertible square matrix for all $k$.*

**Proposition F.3** (Quadratic convergence of Newton-Raphson). *Under the condition of Lemma 3.1, every step iteration $y^{k+1} = y^k - [\nabla D(y^k)^T]^{-1} D(y^k)$. When $D(y)$ is $C_1-$regularity as $\rho_{min}(\nabla D(y)) > C^1$ [7] and satisfying the following condition as*

$$\|\nabla D(y^n) - \nabla D(y^m)\| \leq C_2 \|y^n - y^m\|^2, \quad \text{for some compact sets.}$$

*Thus, the convergence rate is quadratic as $\|y^{k+1} - y^*\| = \mathcal{O}(\|y^k - y^*\|^2)$.*

*Proof.*

$$\|y^{k+1} - y^*\| = \|y^k - [\nabla D(y^k)^T]^{-1} D(y^k) - y^*\| \tag{45}$$

The error gap $D(y^k)$ can be calculated as variational form as:

$$D(y^k) = \int_0^1 \nabla D(y^* + t(y^k - y^*)) dt(y^k - y^*). \tag{46}$$

Plug the equation 46 to equation 45, we get

$$
\begin{aligned}
&\Rightarrow \|y^k - y^* - [\nabla D(y^k)^T]^{-1} D(y^k)\| \\
&= \|[\nabla D(y^k)^T]^{-1} \left[ [\nabla D(y^k)^T](y^k - y^*) - D(y^k) \right] \| \\
&= \|[\nabla D(y^k)^T]^{-1} \left[ [\nabla D(y^k)^T](y^k - y^*) - \int_0^1 \nabla D(y^* + t(y^k - y^*)) dt(y^k - y^*) \right] \| \\
&\leq \|\nabla D(y^k)^T]^{-1}\| \| \int_0^1 [D(y^k)^T] - \nabla D(y^* + t(y^k - y^*)) dt\| \|y^k - y^*\| \\
&\leq C_2 \|\nabla D(y^k)^T]^{-1}\| \|y^k - y^*\|^2 \\
&\leq \frac{C_2}{\gamma} \|y^k - y^*\|^2
\end{aligned}
\tag{47}
$$

where it is easy to see the quadratic convergence relationship $\|y^{k+1} - y^*\| = \mathcal{O}(\|y^k - y^*\|^2)$.

Finally, the proof of quadratic convergence of $\|\tilde{V}_g^{k+1} - V_g^*\| = \mathcal{O}(\|\tilde{V}_g^k - \tilde{V}_g^*\|^2)$ can be a direct result from the equation 47.

Let's give some analysis to connect to the Theorem 3.4 and Corollary F.1. This analysis provides a one-step rollout case. The multi-step rollout case can be extended following the one-step rollout. It should be noted that the multi-step rollout can be understood as a larger step size to make the convergence of the invariant value function. The core idea behind Newton's step is to use the second-order information to guide the convergence of value function $V_g^k(\theta)$. The second-order information of $V_g^k(\theta)$ is from the Bellman optimality $B^*$. By observing equation 23, one-step Temporal Difference is updated by using $r_g(z_t, a_t^*)$ instead of using $r_g(z_t, a_t)$. The $a^*$ derived from provide a piece of second-order information. The value function is defined on the latent space, and a well-learned latent space can boost the convergence of the invariant value function by vector field information. Compared to the conventional RL methods, such as soft actor-critic RL, the policy $\pi$ is updated incrementally, and it is impossible to discover a piece of second-order information to guide the policy.

The proof of the quadratic convergence rate can be analogous to Newton's step. The fact is significantly different from the actor-critic RL methods, where the policy $\pi$ needs to be updated incrementally. The conventional RL relies on asynchronous updating of value function and policy, which causes a low convergence rate (Sutton & Barto, 2018). This paper's analytical form of optimal policy provides "curvature information" to boost the convergence rate.

---

[7] $\rho$ represents the Eigenvalue of matrix.

# G    Pseudo algorithm for KEEC control

---

**Algorithm 2** KEEC: Control

---

**Require:** trained encoder $g_\theta^{\mathrm{en}}$, trained value net $\tilde{V}_g(\cdot, \psi)$ from Algorithm 1;
   reward function $R_1$; optimal state $s^*$; max control time $T_{\max}$; environment $Env$
 1: initial state $s_0 \in M$
 2: $t = 0$
 3: **while** $t \leq T_{\max}$ **do**
 4:    Map to latent space $z_t = g_\theta^{\mathrm{en}}(s_t)$
 5:    perform optimal action $a_t = \pi(z_t)$ from the lifted policy with Equation equation 16 or equation 37
 6:    Observe next state $s_{t+1} = Env(s_t, a_t)$
 7:    $t \leftarrow t + 1$
 8: **end while**

---

## H  Experiment Settings

### H.1  Description of the tasks

**Pendulum.** The swing-up pendulum problem is a classic control problem involving a pendulum swinging from a downward hanging position to an unstable inverted position. In the controlgym-based control, the problem has two state dimensions: angular $\theta$ and angular velocity $\dot{\theta}$. In the image-based control, we simulate the pendulum with the same dynamics, but the state is defined as the binary images of two consecutive states (the $\dot{\theta}$ can be learned by locally differencing). The image has size $96 \times 48$, as the number of pixels (see Figure 6).

The pendulum motion equitation can be expressed as follows:

$$\frac{d^2\theta}{dt^2} = \frac{3g}{2l}\sin\theta + \frac{3}{ml^2}a, \tag{48}$$

where $l$ is the pendulum length, $m$ is mass, $g$ is the gravitational acceleration, and $a$ is the applied torque. Parameter settings commonly used in such studies, i.e., $m = 1, l = 1$, $g = 10$, and $a \in [-2, 2]$, are used in this study. In addition to the results presented in the main context, we show four KKEC control trajectories of the inverted pendulum (gym) with random initials and the 3D visualization of the learned value function in Figure 5.

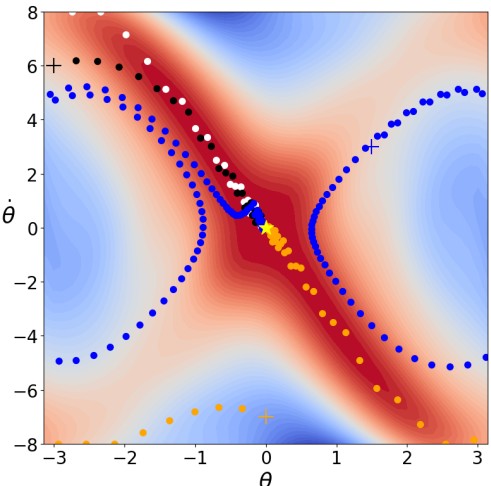

(a) Example control trajectories with four random initials, where + indicates the initials and the yellow star indicates the optimal state. The background contour map indicates the magnitude of the value function (red indicates high value and blue indicates low value)

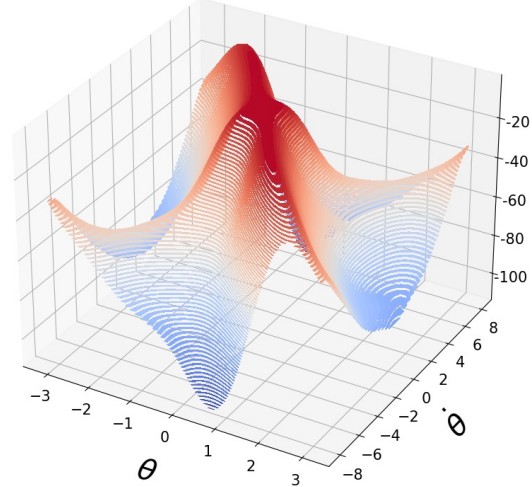

(b) 3D visualization of learned value function

Figure 5: The visualization of the learned value function and the KEEC control trajectories of the swing-up pendulum.

**Lorenz-63 system.** The Lorenz-63 model (Lorenz, 1963; Saltzman, 1962; Yang et al., 2025), which consists of three coupled nonlinear ODEs,

$$\frac{dx}{dt} = \sigma(y - x), \ \frac{dy}{dt} = x(\rho - z) - y, \ \frac{dz}{dt} = xy - \beta z \tag{49}$$

used as a model for describing the motion of a fluid under certain conditions: an incompressible fluid between two plates perpendicular to the direction of the earth's gravitational force. In particular, the equations

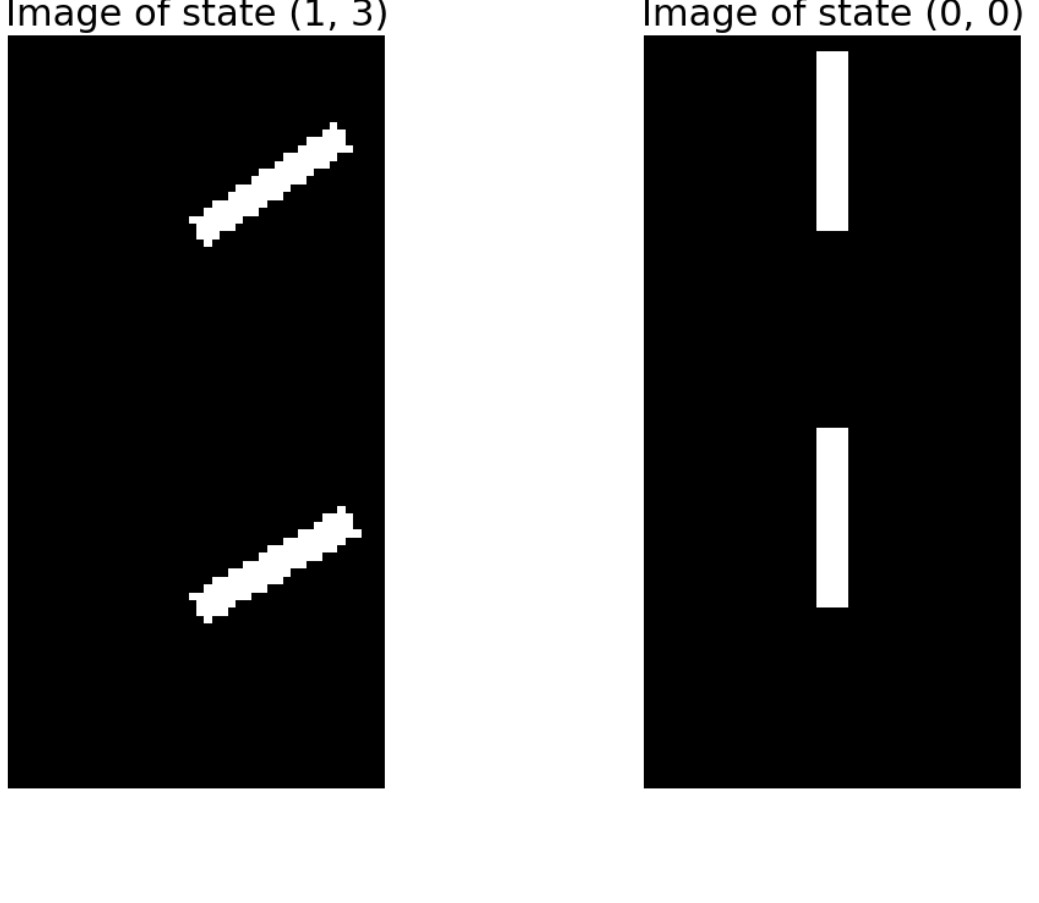

(a) Image of pendulum state $(1, 3)$        (b) Image of pendulum goal state $(0, 0)$

Figure 6: Demonstration of states in the image-based pendulum control task: the image is the concatenated frames from two consecutive states (last timestep and current timestep).

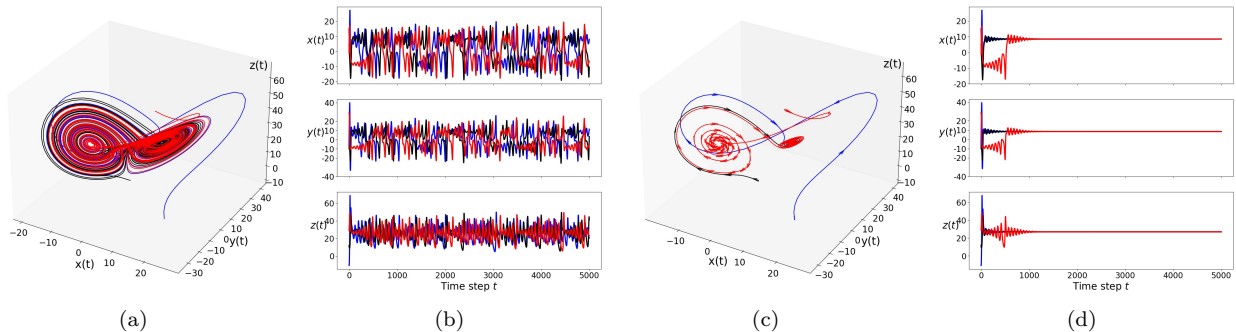

Figure 7: (a) and (c) uncontrolled and KEEC controlled 3d phase trajectories of Lorenz-63, where the arrows in (c) indicate the moving direction. (b) and (d) uncontrolled and controlled state trajectories of Lorenz-63.

describe the rate of change of three quantities with respect to time: $x$ is proportional to the rate of convection, $y$ to the horizontal temperature variation, and $z$ to the vertical temperature variation. The constants $\sigma$, $\rho$, and $\beta$ are system parameters proportional to the Prandtl number, Rayleigh number, and coupling strength. In this paper, we take the classic choices $\sigma = 10$, $\rho = 28$, and $\beta = \frac{8}{3}$ which leads to a chaotic behavior with two strange attractors $(\sqrt{\beta(\rho-1)}, \sqrt{\beta(\rho-1)}, \rho-1)$ and $-(\sqrt{\beta(\rho-1)}, -\sqrt{\beta(\rho-1)}, \rho-1)$. Its state is $s = (x, y, z) \in \mathbb{R}^3$ bounded up and below from $\pm 30$. Our implementation and control settings are based on the (Li et al., 2012) which modifies the Lorenz-63 system with three action inputs $a = (a_x, a_y, a_z) \in [-3, 3]^3$ on each of the variable as,

$$\frac{dx}{dt} = \sigma(y - x) + a_x, \frac{dy}{dt} = x(\rho - z) - y + a_y, \frac{dz}{dt} = xy - \beta z + a_z, \tag{50}$$

The goal is to steer the system's dynamics toward a state $s^* = \mathbf{0}$ with appropriate control inputs. Although the system naturally approaches the attractors, its stabilization is extremely challenging due to fractal oscillation and sensitivity to perturbations. The numerical integration of the system equation 50 using the fourth order Runge-Kutta (Butcher & Wanner, 1996) with a time step $\Delta t = 0.1$. Figure 7 presents four trajectories of the Lorenz-63 system, both uncontrolled and KEEC controlled. This comparison effectively illustrates the effect of KEEC on stabilizing the Lorenz-63 system.

**Wave Equation.** The wave equation is a fundamental second-order PDE in physics and engineering, describing the propagation of various types of waves through a homogeneous medium. The temporal dynamics of the perturbed scalar quantity $u(x, t)$ propagating as a wave through one-dimensional space is given by

$$\frac{\partial^2 u}{\partial t^2} - c^2 \frac{\partial^2 u}{\partial x^2} = a(x, t) \tag{51}$$

where $c$ is a constant representing the wave's speed in the medium, and $a(x, t)$ is a source term that acts as a distributed control force,

$$a(x, t) = \sum_{j=0}^{n_a - 1} \Phi_j(x) a_j(t). \tag{52}$$

The control force consists of $n_a$ control inputs $a_j(t)$, each acting over a specific subset of the spatial domain, defined by its corresponding forcing support function $\Phi_j(x)$ (see Figure 8 for the demonstration of the controller in our implementations). Such a control force can be used to model the addition of energy to the system or other external influences that affect the PDE dynamics. The uncontrolled solution of the wave equation for $c = 0.1$ and initial condition $u(x, t = 0) = \text{sech}(10x - 5)$ can be found in Figure 9. We use the implementation and control environment by the Python package controlgym[8] (Zhang et al., 2023). The wave equation with periodic boundary condition is solved by first transforming (51) into a coupled system of two

---

[8]https://github.com/xiangyuan-zhang/controlgym

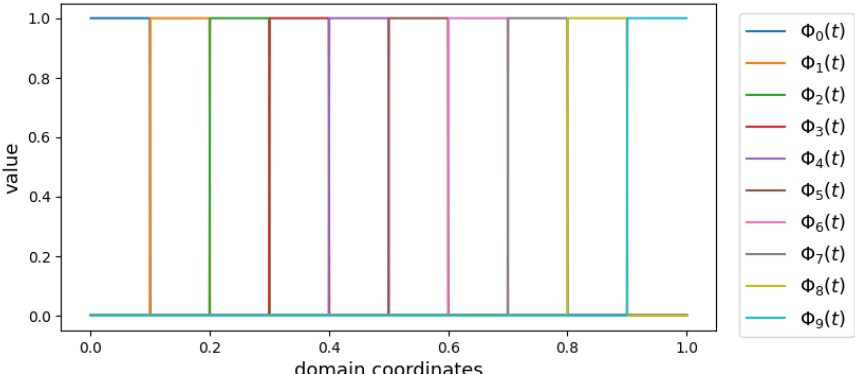

Figure 8: Demonstration of how distributed control inputs influence the dynamics of the wave equation through forcing support functions. In our setting, the forcing support functions corresponding to $n_a = 10$ control inputs are shown, each with a width of 0.1, uniformly affecting the state components of the physical domain.

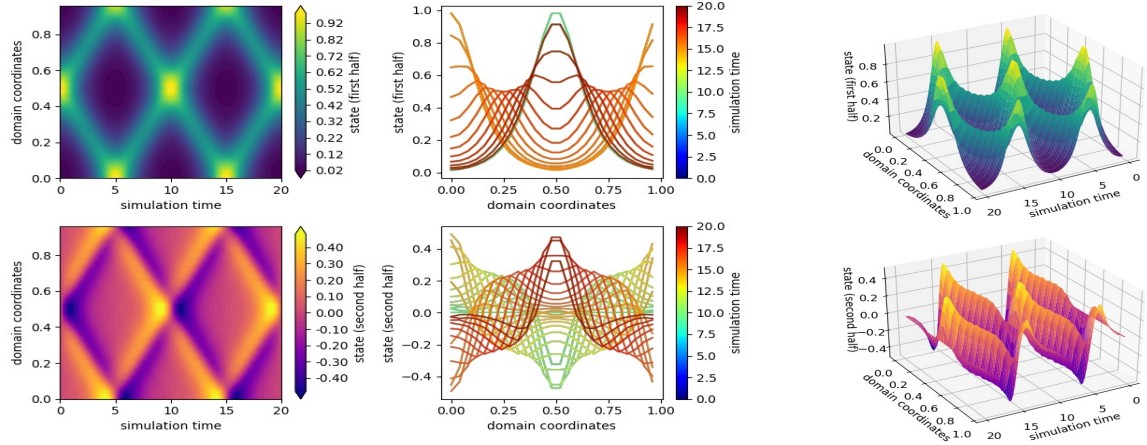

Figure 9: The uncontrolled solution to the wave equation in a spatial domain $[0, 1]$ with parameters, $c = 0.01$. The initial condition is $u(x, t = 0) = \text{sech}(10x - 5)$ and $\psi(x, t = 0) = 0$. (left): Contour plot that shows the value of the state variable over the total simulation time ($x$-axis) and across the spatial domain ($y$-axis). (middle): 1D line representing the state variable at fixed times. The $x$- and $y$-axes represent spatial coordinates and values of the state variable, respectively. The colour of the lines corresponds to the time stamps within the total simulation time. (right): 3D surface plot showing the value of the state variable ($z$-axis) over time ($y$-axis) and across the spatial domain ($x$-axis).

PDEs, with first-order time derivatives defined as $\psi(x, t) = \frac{\partial u}{\partial t}$ representing the rate at which the scalar quantity $u(x, t)$ is changing locally.

In our implementation, we discretized the spatial domain $[0, 1]$ with $\Delta x = 0.02$ and results the discretized field $u_{\Delta x}(x, t) = [u_0, u_{\Delta x}, ..., u_1] \in \mathbb{R}^{25}$. The system state is defined as the combination of discretized $u(x, t)$ and discretized time derivative $\psi(x, t)$ such as $s = [u_0, u_{\Delta x}, ..., u_1, \psi_0, \psi_{\Delta x}, ..., \psi_1] \in \mathbb{R}^{50}$. We set the control term $a(x, t)$ with $n_a = 5$ control inputs $a_j(t) = 1$ with support function $\Phi_j(x) = [0.2j, 0.2(j + 1)]$, as illustrated in Figure 8. We visualize the control trajectories of all the baselines and KEEC in Figures 10, 11, 12, 13, and 14.

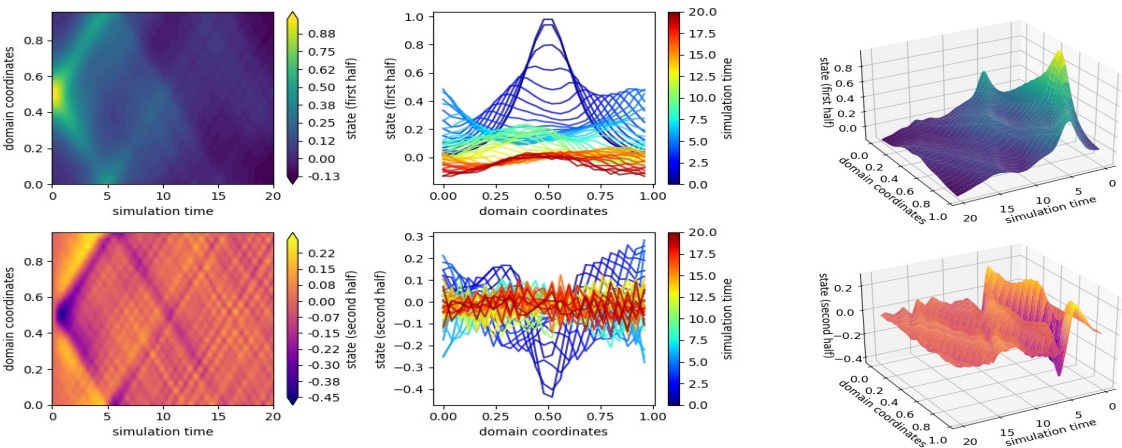

Figure 10: KEEC controlled solution to the wave equation. The figure convention is consistent with Figure 9.

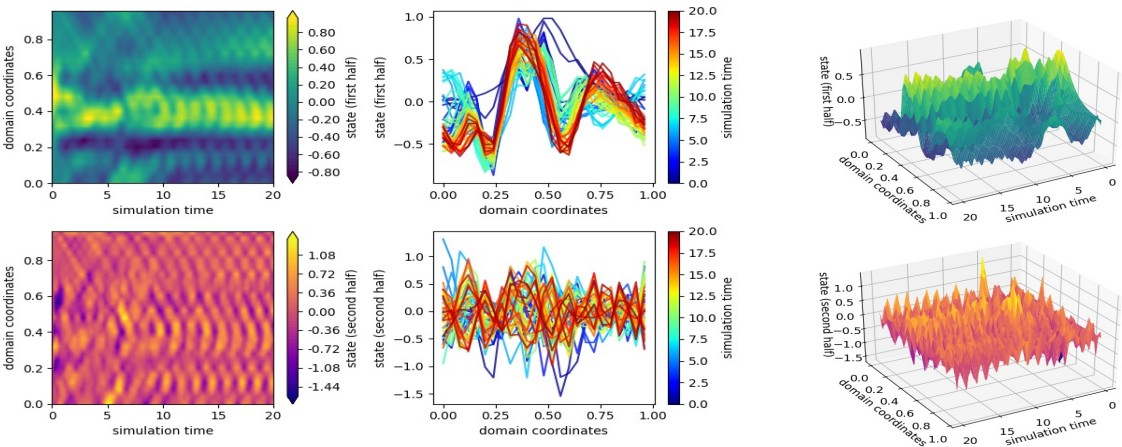

Figure 11: SAC controlled solution to the wave equation. The figure convention is consistent with Figure 9.

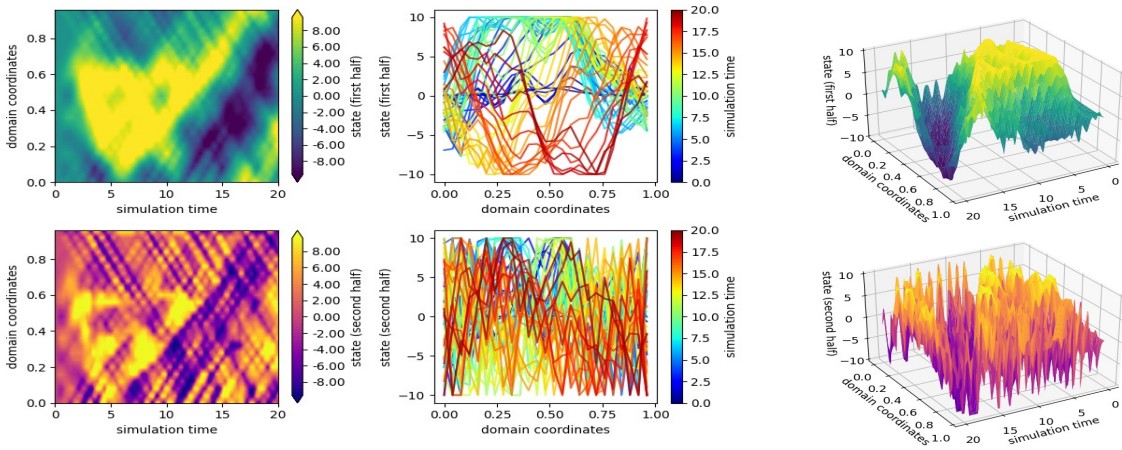

Figure 12: CQL controlled solution to the wave equation. The figure convention is consistent with Figure 9.

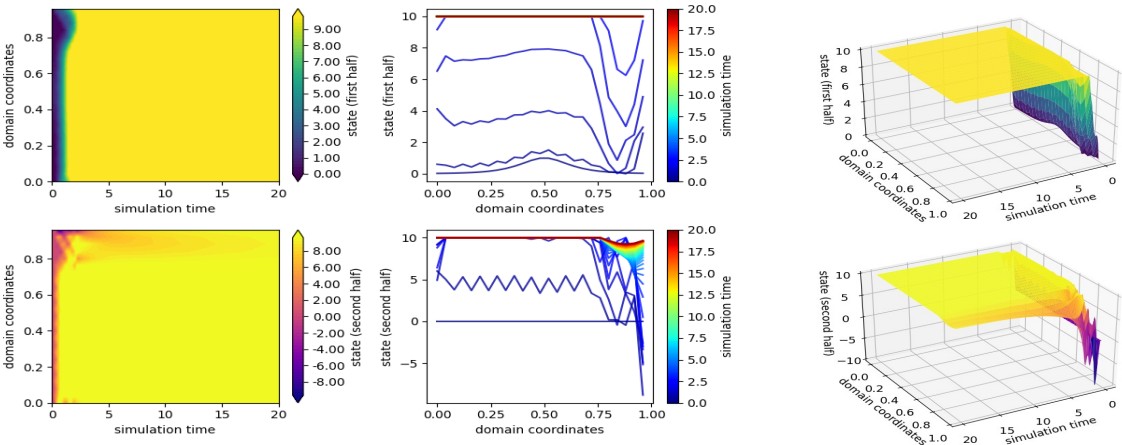

Figure 13: MPPI controlled solution to the wave equation. The figure convention is consistent with Figure 9.

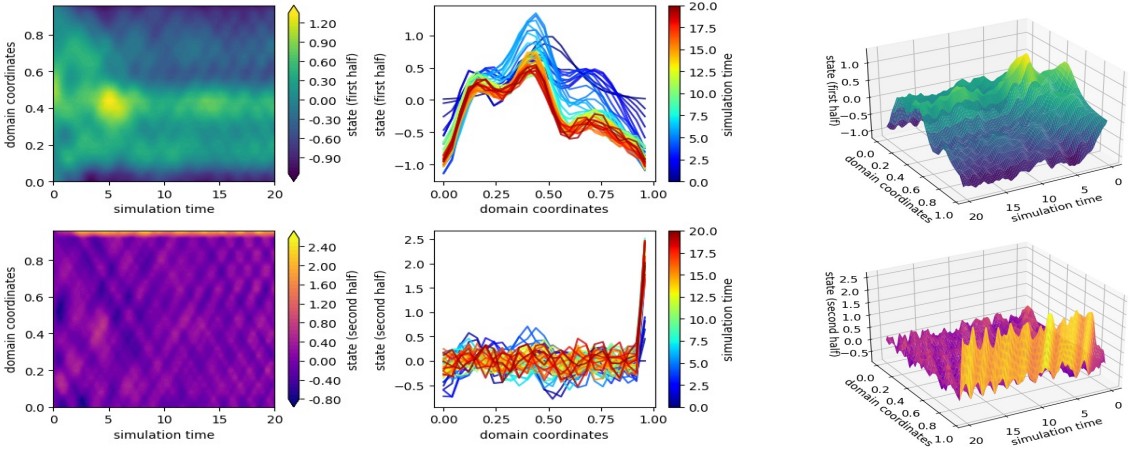

Figure 14: PCC controlled solution to the wave equation. The figure convention is consistent with Figure 9.

## H.2 Training details and baselines.

At a high level, KEEC and other baselines are implemented in **Pytorch** (Paszke et al., 2019). Both training and evaluations are conducted on a Mac studio with a 24-core Apple M2 Ultra CPU and 64-core Metal Performance Shaders (MPS) GPU. The evaluation is conducted on the CPU.

All models were trained using the Adam optimizer (Kingma & Ba, 2015), with the decaying learning rate initially set to 0.001. For KEEC, 100 training epochs are used for system identification with a batch size of 128, and 50 training epochs are used for learning the value function with a batch size of 256. The latent space dimensions are set at 8, 8, 16, and 64 for the pendulum (gym, image) and Lorenz-63, wave equation tasks, respectively. We set the lose weight $\lambda_{\mathrm{met}} = 0.3$. Both offline models (CQL, MPPI, and PCC) were trained for 100 epochs with a batch size of 256, while the online SAC was updated every 10 steps with the same batch size and a 100k replay-memory buffer for a total of 250k gradient iterations. We use the following Pytorch (Paszke et al., 2019) implementations for the baselines. For all the baselines, we keep the default model and hyper-parameter settings, adapting only the state-action dimensions for each task. Additionally, the hidden dimensions of the models are set to be the same as the latent space dimension of KEEC.

- SAC: Stable Baselines 3 (Hill et al., 2018):`https://github.com/hill-a/stable-baselines`.

- CQL: The implementation is based on the provided code of (Kumar et al., 2020):`https://github.com/aviralkumar2907/CQL`.

- MPPI: The implementation is based on the provided code of (Williams et al., 2017):`https://github.com/UM-ARM-Lab/pytorch_mppi`.

- PCC: The implementation is based on the provided code of (Levine et al., 2019): `https://github.com/VinAIResearch/PCC-pytorch`.

## H.3 Evaluation details

The table 2 shows the general settings for the conducted evaluations.

Table 2: Evaluation settings.

|  | initial region | goal state | horizon | noises |
|---|---|---|---|---|
| pendulum (gym) | $[-2.9, -\pi] \times [-8, 8]$ | $(0, 0)$ | 100 | N/A |
| pendulum (image) | $[-2.9, -\pi] \times [0]$ | $(0, 0)$ | 100 | N/A |
| Lorenz-63 | $[-1, -17, -20] \pm [1, 1, 1]$ | $(-8, -8, 27)$ | 500 | N/A |
| Wave equation | $u(x, 0) = \mathrm{sech}(10x - 5), \psi(x, 0) = 0$ | $\mathbf{0} \in \mathbb{R}^{50}$ | 200 | $\mathcal{N}(0, 10^{-2})$ |

The SAC, CQL, and KEEC are standard feedback control algorithms that determine control actions based on the observed state. Below, we provide additional settings for two other baseline methods, MPPI and PCC:

- MPPI: The planning horizons are set to 10, 10, 50, and 20 for the pendulum (gym, image), Lorenz-63, and wave equation tasks, respectively. The number of sampled trajectories for integral evaluations is fixed at 100 across all tasks.

- PCC: This method utilizes the iLQR algorithm to perform control in its latent space. For the two pendulum tasks, we keep the default settings as outlined in the original paper. For the Lorenz-63 and wave equation tasks, the latent cost matrices are set to match the cost matrices $R_1$ and $R_2$ in the original spaces. The planning horizons are set to 10 for each pendulum task, 50 for Lorenz-63, and 20 for the wave equation. The number of iLQR iterations is consistently set at 5 for all tasks.

### H.4 Model Architecture

In the implementations, an autoencoding architecture is used for Koopman embedding, where the encoder and decoder are symmetric and contain only three Fully Connected (FC) layers each. The performance of the proposed KEEC can be easily verified using a simple design and demonstrates its potential to solve more complex control tasks with more well-designed neural networks. It is recalled that $m, d$ represents the dimension of the environmental state and input control signal, respectively, whereas $n$ is the dimension of latent space (i.e. the finite approximated dimension of our Koopman operators). For the latent value function, two types of model architectures were employed: (1) Multi-Layer Perception (MLP) and (2) Quadratic Form $V_g(z) = (z - z^*)^T W(z)(z - z^*)$ where $W(z) = W^{\frac{1}{2}}(z)^T W^{\frac{1}{2}}(z)$ ensures the positive definiteness and $z^*$ is the encoded optimal state $s^*$. Specifically, the general network structure is listed in Table 3, including the specific sizes used and the different activation functions.

Table 3: KEEC model architecture in our implementation

| Components | Layer | Weight Size | Bias Size | Activation Function |
|---|---|---|---|---|
| Encoder | FC | $m \times \frac{n}{2}$ | $\frac{n}{2}$ | Tanh |
| Encoder | FC | $\frac{n}{2} \times n$ | $n$ | Tanh |
| Encoder | FC | $n \times n$ | $n$ | Tanh |
| Encoder | FC | $n \times n$ | $n$ | None |
| Decoder | FC | $n \times n$ | $n$ | Tanh |
| Decoder | FC | $n \times \frac{n}{2}$ | $\frac{n}{2}$ | Tanh |
| Decoder | FC | $\frac{n}{2} \times m$ | $m$ | Tanh |
| Decoder | FC | $\frac{n}{2} \times m$ | $m$ | None |
| Value Function (MLP) | FC | $n \times n$ | $n$ | ReLU |
| Value Function (MLP) | FC | $n \times \frac{n}{2}$ | $\frac{n}{2}$ | ReLU |
| Value Function (MLP) | FC | $\frac{n}{2} \times \frac{n}{2}$ | $\frac{d}{2}$ | ReLU |
| Value Function (MLP) | FC | $\frac{n}{2} \times 1$ | $1$ | None |
| Value Function (Quadratic) | FC | $n \times (n \times n)$ | $n \times n$ | None |

As discussed in the main text, no extra parameters are used for training the two lifted $\mathcal{P}$ and $\mathcal{U}$ instead of solving the least square minimization problem (8) with an analytical solution to obtain $\hat{\mathcal{P}}$ and $\hat{\mathcal{U}}$ with regularization $10^{-3}$. The solved solutions in each batch were averaged over all the training data. In evaluation, the two approximated lifted operators $\hat{\mathcal{P}} \in \mathbb{R}^{n \times n}$ and $\hat{\mathcal{U}} \in \mathbb{R}^{n \times d}$ are loaded into the dynamics model and used in the control tasks.

