# OpenReview forum: "Koopman Embedded Equivariant Control"
_TMLR — Rejected by TMLR_

### Review · Reviewer_VpVC · 2025-07-15

**Summary Of Contributions:**

This paper introduces Koopman Embedded Equivariant Control (KEEC), a method that learns a latent representation of system states and their associated vector fields, including state derivatives. The approach ensures control consistency by enforcing equivariance and isometry in the embedding space. Leveraging the linear structure of the Koopman operator, KEEC facilitates efficient learning of latent dynamics and yields an analytically tractable greedy control policy. Numerical experiments validate the efficacy of the proposed algorithm.

**Audience:**

Yes

**Broader Impact Concerns:**

N.A.

**Claims And Evidence:**

Yes

**Requested Changes:**

* Theoretical Clarity: The paper would benefit from a more detailed explanation of the theoretical results (e.g., Theorem 3.4), including their implications and significance for the proposed method. The provided discussion is too brief to properly assess the contribution of these theoretical guarantees.
* Technical Novelty: The authors should more clearly differentiate their approach from prior work. A explicit comparison with existing Koopman-based or equivariant control methods would help readers understand the advancements made by KEEC.
* Loss Function Justification: The motivation behind using the two distinct loss functions (e.g., their roles in enforcing equivariance, isometry, or control consistency) needs to be explained more thoroughly.
* Writing Quality: There are several typographical errors (e.g., "equation 16" on page 7 is incorrectly referenced) and instances of unclear phrasing. A thorough proofreading to improve grammar, clarity, and technical precision is recommended.

**Strengths And Weaknesses:**

Strengths:

* The paper is well-structured and presents its ideas in a clear, logical progression.
* The proposed algorithm effectively constructs equivariant flows and vector fields, demonstrating sound theoretical foundations.
* The numerical experiments are comprehensive, which includes larger-scale examples.

Weaknesses:

* The technical contribution is unclear compared to existing methods in Koopman operator-based control and equivariant systems.
* The method's performance appears heavily dependent on sample size due to its requirement for small time steps, which may limit practical applications.

---

> ### Author Response · Authors · 2025-08-22
> **Rebuttal-1**
>
> We sincerely thank the reviewer for the careful reading of our paper and the constructive feedback. We greatly appreciate the recognition of our paper’s clarity, theoretical foundation, and comprehensive experiments. Below, we provide point-by-point responses to the concerns.
>
>
>
> **Weakness 1.** `The technical contribution is unclear.`
>
> **Answer.** Our work addresses the fundamental question: `What constitutes an effective latent embedding $g$ for controlling systems with unknown nonlinear dynamics?`
>
> Our contributions can be summarized in two parts, spanning a new theoretical framework and a novel algorithm.
>
> 1. We define a "Comprehensive Embedding" framework for control, ensuring the consistent control performance in original and latent spaces.
>
>     - **Our approach to equivariance targets the dynamics' implicit symmetries, distinguishing it from standard methods.**  While many existing works enforce equivariance to _explicit geometric symmetries_ of a system (e.g., SO(2) or SO(3) rotations), our method learns an equivariant mapping for the _entire manifold structure_ itself. This establishes an equivariance between the nonlinear flows and vector fields on the original state manifold and the learned linear evolution in the latent embedding space. This comprehensive mapping of the system's underlying differential equations, formally justified by the isomorphism between the Lie group (local flows) and the Lie algebra (local vector fields) (as detailed in Section 3.2 and Appendix D).
>
>     - **We formally propose that this flow equivariance, when combined with isometry, is the sufficient condition for preserving control consistency.** Isometry ensures that the metric structure is preserved between the original and latent spaces. This is critical for learning an invariant value function (Lemma 3.3) and ensuring that an optimal policy in the latent space remains effective when mapped back to the original system. We operationalize this principle through a novel isometry loss function, $\mathcal{E}_{met}$, which explicitly minimizes geometric distortion and addresses a key deficiency in prior embedding methods.
>
> 2. We propose a novel model-based RL algorithm that leverages the comprehensive embedding (including equivariant flow and vector field) to derive a more robust, analytical control policy.
>
>     - Building on our geometric control theory, our algorithm allows the infinitesimal generators $\mathcal{P, U}$ to directly embed and linearize the continuous-time equivariant flows of the controlled dynamics (see the discussion in Section 3.3).
>
>     - This learned differential information (based on comprehensive embedding) is then used to derive an analytical form of the greedy control policy via Hamiltonian-Jacobi theory (Equations 15 and 16). This approach avoids the need for costly iterative numerical optimization at each control step and value iteration, leading to significant gains in computational efficiency (Figure 3). Furthermore, the structure of our framework provides strong convergence guarantees for the value function (Proposition F.3 in Appendix F.3), which contributes to the overall stable learning process for value iteration.

---

> ### Author Response · Authors · 2025-08-22
> **Rebuttal-2**
>
> **Weakness 2.** `The method's performance appears heavily dependent on sample size due to its requirement for small time steps, which may limit practical applications.`
>
> ***Answer**.  We thank the reviewer for raising this important point regarding practical applications. We wish to clarify that this requirement is common in the field, and does not substantially limit applicability in many domains.
>
> - **The Required Time Step t = 0.1s is Consistent with Common Practice in This Field.** This step size is widely adopted in the literature for dynamical system simulation and model-based control. Many state-of-the-art studies employ comparable or even smaller time steps to ensure accurate modeling [1,2,3]. More specifically, our paper uses a time step of $t = 0.1s$ (see the paragraph under Equation (50)), whereas paper [4] adopts $t = 0.01s$ (see the paragraph under Equation (26)), which is ten times smaller.
>
>
>
> - **The Dependency is on Sampling Frequency, Not Necessarily Data Volume.** Our method requires high temporal resolution within trajectories but does not necessarily require a larger volume of diverse training scenarios compared to other model-based methods.
>
> - **High-Frequency Data is Available in Many Practical Domains.** Crucially, this requirement does not limit the method's practical applications in many key areas. High-frequency data is common and often essential in domains such as robotic manipulation, autonomous driving, and the control of PDE systems, where fast and accurate state estimation is critical.
>
>
>
> [1] Dawson, Charles, et al. "Safe nonlinear control using robust neural lyapunov-barrier functions." Conference on Robot Learning. PMLR, 2022.
>
> [2] Qin, Zengyi, et al. "Learning safe multi-agent control with decentralized neural barrier certificates." arXiv preprint arXiv:2101.05436 (2021).
>
> [3] Yildiz, Cagatay, Markus Heinonen, and Harri Lähdesmäki. "Continuous-time model-based reinforcement learning." International Conference on Machine Learning. PMLR, 2021.
>
> [4] V. Zinage & E. Bakolas, Neural Koopman Lyapunov Control, Neurocomputing, 2023. (github: https://github.com/Vrushabh27/Neural-Koopman-Lyapunov-Control/blob/main/Simple_Pendulum.ipynb)
>
>
>
> **Request to change 1.** `Theoretical Clarity: The paper would benefit from a more detailed explanation of the theoretical results (e.g., Theorem 3.4), including their implications and significance for the proposed method. The provided discussion is too brief to properly assess the contribution of these theoretical guarantees.`
>
>
> **Answer.** Thank you for pointing this out. We have revised the manuscript to provide a clearer explanation after Theorem 3.4, including its implications and significance. The changes can be summarized as follows:
>
> 1. **Clarification of implication.** We now explain that the analytical solution in Equation (16) gives the optimal control policy in the latent space and directly corresponds to the maximization problem in the Hamilton–Jacobi Equation (15).
>
> 2. **Methodological significance.** Because we introduce the invariant value function together with the equivariant vector fields, the resulting policy is derived in a consistent manner directly from these learned structures. This removes the need to train an additional policy neural network.
>
> 3. **Theoretical benefit.** The obtained analytical control policy can be directly incorporated into the Bellman value function iteration in Equation (14). This integration provides a strong convergence guarantee toward the optimal invariant value function (see Appendix B.3).
>
> 4. **Practical benefit.** The closed-form policy significantly reduces the time required for generating optimal control actions, as demonstrated in Figure 3 (left).
>
> These revisions make the technical novelty of KEEC and its distinction from prior work much clearer.

---

> > ### Author Response · Authors · 2025-08-22
> > **Rebuttal-3**
> >
> > **Request to change 2.** `Technical Novelty: The authors should more clearly differentiate their approach from prior work. A explicit comparison with existing Koopman-based or equivariant control methods would help readers understand the advancements made by KEEC.`
> >
> >
> > **Answer.** Thank you for this valuable comment. We have revised the introduction and related work sections to more clearly highlight the technical novelty of KEEC compared to prior approaches:
> >
> > 1. **Difference from existing Koopman-based methods.** Prior works primarily embed the states into a latent space. However, state-only embeddings are insufficient to capture the full manifold structures required for control. Unlike existing Koopman-based methods that only embed states, KEEC leverages the natural correspondence between Lie groups (flows) and Lie algebras (vector fields) to embed both into the latent space (see the discussion in Section 3.2 and Appendix D pp. 21). This geometric construction ensures equivariance and preserves the full structure of the entire manifold. Consequently, KEEC establishes that preserving equivariance is essential for a valid embedding, ensuring that the latent representation faithfully retains the control structure of the original system.
> >
> > 2. **Difference from existing equivariant control methods.** We answer this questions from the following perspectives
> >
> >     - **Different Aims.** Most equivariant control methods aim to exploit symmetries to improve sample efficiency, and they typically enforce equivariance with respect to explicit geometric symmetries of a system (e.g., groups $E(3)$, $SO(2)$ or $SO(3)$ rotations). Conversely, our methods is centered on addressing the question _"what constitutes an appropriate embedding for control?"_. Here, equivariance is treated as a **necessary** condition for constructing such an embedding, rather than simply a tool for exploiting symmetry.
> >
> >     - **Different Methods.** In most equivariant control methods, explicit geometric groups are manually encoded into the loss function to enforce equivariance. By contrast, KEEC enforces equivariance by learning the underlying Lie group (flows) and Lie algebra (vector fields) on a smooth manifold, enabling a comprehensive embedding of flows and vector fields.
> >
> > 3. **Advancement of KEEC.** Because our embedding satisfies two necessary conditions—equivariance and isometry—we can derive an invariant value function that guarantees consistent control performance in both the original and latent spaces. Building on this invariant value function and the learned equivariant vector fields, KEEC admits an analytical control policy without the need for training a separate policy network. This design ensures strong convergence of value iteration and significantly improves the efficiency of policy generation.
> >
> > Thank you for the comment. We have revised the main text to incorporate your suggestions, clarifying KEEC’s contributions and strengthening the comparison with existing methods.

---

> ### Author Response · Authors · 2025-08-22
> **Rebuttal-4**
>
> **Request to change 3. Loss Function Justification:.** `The motivation behind using the two distinct loss functions (e.g., their roles in enforcing equivariance, isometry, or control consistency) needs to be explained more thoroughly.`
>
> **Answer.** Thank you for raising this point. We have revised the manuscript to clarify the motivation and the precise role of our two distinct loss functions:
>
> 1. **The Role of the Forward Loss $\mathcal{E}_{fwd}$: Enforcing Equivariance and Accurate Representation.**
>
>     This loss, defined in Equation (10), combines two essential terms. Its primary role is to enforce that our learned latent dynamics are an equivariant map of the true system's evolution.
>
>     - **The correction of the equivariant flow:** This term enforces equivariance by penalizing the error between the true next state $s_{t+\Delta t}$ and the state predicted by our latent dynamics model. It is implemented by using the approximated infinitesimal generators $\mathcal{P}$ and $\mathcal{U}$ to predict the latent flow: $    z_{\Delta t} \approx \exp(\mathcal{P}\Delta t) z_0 + \mathcal{P}^{-1} \left(\exp(\mathcal{P}\Delta t) - I\right)\mathcal{U}(z_0)a_0.$
>     In this way, both flows and vector fields are simultaneously captured under the embedding $g$.
>
>
>     - **The reconstruction term**  This loss guarantees that the embedding itself is a faithful representation of the state.
>
> 2. **The Role of the Isometry Loss $\mathcal{E}_{met}$: Enforcing a Consistent Metric for Control.**
>
>     We emphasize that control tasks are highly dependent on metric information, as reward functions are often defined by distances (e.g., distance to a goal state). If the metric is distorted during embedding, a policy that is optimal in the latent space may fail when transferred back to the original space.
>
>     The isometry loss directly addresses this by explicitly penalizing any geometric distortion between the two spaces. This ensures that the metric structure is preserved, which ensures consistent control performance across both spaces and the invariance of the value function (see Lemma 3.3).
>
>
> Together, these two losses enforce the necessary conditions for constructing a valid control embedding: equivariant loss ensures the dynamic evolution is mapped correctly (equivariance), while metric loss ensures the geometric structure is preserved (isometry). Building on this foundation, KEEC can learn an invariant value function and derive an analytical control policy with consistent optimal performance.
>
>
> **Request to change 4. Writing Quality.**  `There are several typographical errors (e.g., "equation 16" on page 7 is incorrectly referenced) and instances of unclear phrasing. A thorough proofreading to improve grammar, clarity, and technical precision is recommended.`
>
> **Answer.** Thank you for pointing this out. We have carefully proofread the manuscript, corrected the incorrect reference to “equation 16” on page 7, and revised several sentences to improve grammar, clarity, and technical precision.
>
> --------------------
> We thank the reviewer once again for these valuable comments. We have revised the manuscript accordingly by clarifying the theoretical results, explicitly differentiating KEEC from prior work, strengthening the justification of the loss functions, and carefully proofreading for improved clarity and accuracy. We believe these revisions address the concerns raised and further highlight the novelty and contributions of our work.

---

### Review · Reviewer_DVyM · 2025-08-08

**Summary Of Contributions:**

In this article, the authors introduce Koopman Embedded Equivariant Control (KEEC), a control framework for nonlinear control-affine systems with unknown dynamics, grounded in the Koopman operator theory. Unlike existing approaches, KEEC aims to learn a mapping from the original state space to a latent space that is both isometric and equivariant, thereby preserving the control effects across both spaces. This mapping is learned through an autoencoder architecture, with additional terms in the loss function specifically promoting the isometric and equivariant properties. The resulting latent space dynamics are linearized, allowing for the derivation of an optimal controller using Hamilton-Jacobi theory. The learned value function is then used to control the original nonlinear system.

The main contribution lies in incorporating isometric and equivariant mappings into the Koopman framework, enabling a structured and interpretable controller design.

**Audience:**

Yes

**Claims And Evidence:**

Yes

**Requested Changes:**

- Conduct a more comprehensive literature review, with a focus on recent data-driven Koopman frameworks.

- Clearly highlight the benefits of the isometric and equivariant mappings, both in the text and through empirical comparisons with other state-of-the-art methods.

**Strengths And Weaknesses:**

Strengths:

- The article builds on the promising data-driven Koopman framework, which has shown strong potential in recent years for designing controllers for nonlinear systems with unknown dynamics.

- The inclusion of isometric and equivariant mappings is well-motivated, particularly from the perspective of interpretability and ensuring physically meaningful embeddings.

- The article is clearly written, and the mathematical derivations appear sound.

Weaknesses:

- Since the core contribution lies in incorporating isometric and equivariant mappings into the framework, their practical benefit should be more convincingly demonstrated. Rather than only comparing KEEC to other control approaches, it would be valuable to show the impact of not including these properties—e.g., ablation studies removing the isometric, equivariant, or both terms from the VAE cost function. Additional strategies could be explored to empirically highlight the specific advantages of these properties beyond their theoretical appeal.

- The literature review largely references older works. However, there has been a surge of recent interest in data-driven Koopman methods, especially within the control community. For instance, the recent CDC 2024 workshop on data-driven Koopman methods (https://www.tu-ilmenau.de/cdc24), including work from Sandra Hirche’s group, presents a number of relevant advances. It would strengthen the paper to position KEEC within this more current context and compare it to these recent methods.

- After Proposition 3.2: While Equation (7) is understood, the subsequent sentence refers to it as an approximation of the next latent space, which could use clarification—what exactly is being approximated here?

Minor issues:

- Section 3.2: “derivative. equation 5 shows that” → “derivative. Equation 5 shows that”

- Section 3.3: “Instead, deep learning, which is well-suited for representing arbitrary functions.” – This sentence appears incomplete; a verb or concluding clause is likely missing.

---

> ### Author Response · Authors · 2025-08-22
> **Rebuttal-1**
>
> We sincerely thank the reviewer for the thoughtful and constructive feedback. We greatly appreciate the recognition of our contributions and the helpful suggestions regarding literature coverage, empirical validation of isometric and equivariant mappings, and clarification of technical details. Below we provide point-by-point responses to each of the raised comments.
>
>
> **Weakness 1** `Since the core contribution lies in incorporating isometric and equivariant mappings into the framework, their practical benefit should be more convincingly demonstrated.`
>
> **Answer**. Thank you for this constructive feedback. We agree that clearly demonstrating the practical benefits of these properties is crucial.
>
> ***On the Ablation Studies***
>
> We would like to clarify that an ablation study for the isometric loss is included in the manuscript.
>
> 1. Quantitative Results: As shown in Table 1, the KEEC (w/o $\mathcal{E}_{met}$) variant, which removes the isometric loss, fails to achieve successful control, demonstrating that this property is essential for the performing control in the latent space.
>
> 2. Qualitative Results. Figure 4 provides a visual counterpart to this finding. With the isometry loss, the learned latent space remains smooth and preserves the system’s metric structure (see Subfigure (b)), whereas without it the space becomes distorted and an optimal trajectory cannot be identified (see Subfigure (d)). Moreover, in the ablation study (Section 4.1, Figure 3d), we vary the parameter $\lambda_{met}$, defined in Equation (10) as the weight of the isometry constraint, over the interval $[0.1,1]$. The results show that KEEC keeps robust when $\lambda_{met} \in [0.1,0.3]$
>
> We have revised our manuscript to more explicitly highlight these results as a direct demonstration of the effectiveness of isometric loss term.
>
> ***On the VAE Comparison***
>
> Thanks for your suggestion to compare with a VAE. As recognized by the reviewer, an ablation removing both our isometric and equivariant terms would effectively reduce our model to a _deterministic_ autoencoder for learning latent space. Our existing ablation (KEEC (w/o $\mathcal{E}_{met}$)) already shows that removing just one of these key structural losses causes the control task to fail. This implies that a baseline without any of these priors would perform even more poorly. We chose not to compare against a standard probabilistic VAE for two key reasons:
>
> 1. Our method handles deterministic dynamics, and introducing the stochasticity from a VAE's KL-divergence term would be misaligned with the property.
>
> 2. Our model is designed to simultaneously learn a structured latent space and its dynamics, whereas a standard VAE's objective, focused on reconstruction and prior matching (KL loss), deviates from this purpose and our algorithm design.
>
>
>
> **Weakness 2.** `The literature review largely references older works.`
>
> **Answer**. Thank you for the constructive feedback. We agree that it is important to incorporate recent developments in data-driven Koopman operator theory. Accordingly, we have added the following paragraph to the Introduction of revised manuscript:
>
>     > An additional line of research on embeddings for control involves Koopman operator theory. The Koopman operator was originally proposed by Koopman and von Neumann as a linear embedding for Hamiltonian dynamical systems (Koopman, 1931; Koopman & Neumann, 1932). However, due to its infinite-dimensional nature, identifying suitable handcrafted basis functions is challenging with conventional methods (Brunton et al., 2021). The advent of deep learning has greatly advanced this area by enabling data-driven discovery of effective representations of dynamical systems. For instance, extended dynamic mode decomposition (E-DMD) has been introduced as a linear approximation of the Koopman operator (Proctor et al., 2018). Building on this, nonlinear basis functions such as kernel methods and autoencoders have also been explored for system identification in general nonlinear dynamics (Lusch et al., 2018; Bevanda et al., 2024a;b; 2025, Wi et al., 2025). Thanks to its linearity and universal representational capacity, the Koopman operator has attracted significant attention in control tasks. A representative example is the integration of Koopman embedding with model predictive control (Korda & Mezić, 2020). In addition, model-based reinforcement learning (RL) leverages the Koopman operator as an offline model to improve sample efficiency (Weissenbacher et al., 2022). Beyond optimal control tasks, the Koopman operator has also been employed with certificate functions such as Lyapunov functions and control barrier functions to address safety and stability issues (Folkestad et al., 2020; Zinage & Bakolas, 2023).
>
> This paragraph has been incorporated into the latest version of the manuscript.

---

> ### Author Response · Authors · 2025-08-22
> **Rebuttal-2**
>
> Weakness 3 `After Proposition 3.2: While Equation (7) is understood, the subsequent sentence refers to it as an approximation of the next latent space, which could use clarification—what exactly is being approximated here?`
>
> **Answer.**  Thank you for this helpful comment. We agree that clarification is needed. In Equation (7), the equivariant flow representation
>
> $z_{\Delta t} = \exp(\mathcal{P}\Delta t) z_0 + \int_{0}^{\Delta t} \exp(\mathcal{P}[\Delta t -t]) \mathcal{U}(z_t)a_t  dt$
>
> is exact. However, to make it computationally tractable, we approximate the integral by replacing $\mathcal{U}(z_t)$ with its value at the initial time, $\mathcal{U}(z_0)$ in a short time interval $[0, \Delta t]$. This yields
>
> $z_{\Delta t} \approx \exp(\mathcal{P}\Delta t) z_0 + \mathcal{P}^{-1}\left(\exp(\mathcal{P}\Delta t) - I\right)\mathcal{U}(z_0)a_0,$
>
> which is the expression we present in the proof of Proposition 3.2. The approximation error arises precisely from substituting $\mathcal{U}(z_0)$ for the time-dependent $\mathcal{U}(z_t)$. Thus, Equation (7) can be understood as a tractable approximation of the exact equivariant flow, obtained by approximating the integral with $\mathcal{U}(z_0)$.
>
>
>
>
> **Minor Issues 1.** `Section 3.2: “derivative. equation 5 shows that” → “derivative. Equation 5 shows that”`
>
> **Answer.**  Thank you for pointing this out. The typo has been corrected in the revised manuscript.
>
> **Minor Issues 2.** `Section 3.3: “Instead, deep learning, which is well-suited for representing arbitrary functions.” – This sentence appears incomplete; a verb or concluding clause is likely missing.`
>
> **Answer.**  Thank you for pointing out this issue. We agree that the original sentence was incomplete. We have revised it to make the expression complete and more precise. The sentence now reads:
>
>     Instead, we leverage deep learning, which is well-suited for general-purpose function approximation. In our work, we employ a deep auto-encoder to learn embeddings that naturally align with the Koopman framework.
>
> This revision resolves the grammatical incompleteness and clarifies the role of deep learning for construction of latent representation.
>
> **Request to change 1.** `Conduct a more comprehensive literature review, with a focus on recent data-driven Koopman frameworks.`
>
> The answer refers to `Weakness 2`.

---

> ### Author Response · Authors · 2025-08-22
> **Rebuttal-3**
>
> **Request to change 2.** `Clearly highlight the benefits of the isometric and equivariant mappings, both in the text and through empirical comparisons with other state-of-the-art methods.`
>
>
> We appreciate this valuable suggestion. In the revised manuscript, we have made the benefits of isometric and equivariant mappings more explicit. Specifically, we highlight that:
>
>     From a geometric perspective, our work addresses a central question in latent embedding: what constitutes an appropriate embedding for control? To this end, we formally propose that equivariance and isometry are two essential properties that must be preserved in the latent space to ensure effective control. Based on manifold and Lie theory (Omori, 2017; Lie, 1893), we show that the Koopman operator enables a comprehensive embedding of both equivariant flows and vector fields, thereby preserving the full structure of the dynamics. By incorporating isometry, the invariant value function can be derived, which preserves consistent optimal control performance in both the original and embedded spaces. Leveraging this invariant value function and comprehensive embedding, we further obtain an analytical control policy via Hamiltonian–Jacobi theory, without the need to train a separate policy neural network. This result establishes a convergence toward the optimal value function and enables efficient policy generation.
>
> In addition to clarifying these advantages, we now explicitly discuss how these properties translate into practical benefits for control (e.g., invariant value function, policy efficiency, and value function convergence). More detailed discussion has been incorporated into the latest version of the manuscript.
>
> ---------------------
> We thank the reviewer once again for the valuable comments. We have revised the manuscript accordingly by expanding the literature review, clarifying the theoretical approximations, and adding ablation studies to highlight the benefits of isometric and equivariant embeddings. We believe these revisions address the concerns raised and further strengthen the clarity and contribution of our work.

---

### Review · Reviewer_vvzJ · 2025-08-13

**Summary Of Contributions:**

The paper presents an approach to design a control policy for system with unknown dynamics based on Koopman operator theory

**Audience:**

Yes

**Claims And Evidence:**

Yes

**Requested Changes:**

NA

**Strengths And Weaknesses:**

The paper presents an approach to design a control policy for system with unknown dynamics based on Koopman operator theory. I have extensively worked on Koopman operator and the following are some of my reviews:
1) There have already been extensive works on the problem statement discussed in this paper. In addition, their algorithms have been applied to complex high dimensional robotic systems
2) This paper compares their algorithm with classical yet simpler nonlinear systems that have already been well studied in literature.
3) Furthermore, the authors have not compared their approach with existing Koopman based control approaches (please see [1]. [2], [3], [4] and the references therein)
4) The loss functions considered in this paper such as equations (8), (9), (10) and (11) have already been considered in several papers in the research literature


[1] Vaidya, Umesh. "When Koopman Meets Hamilton and Jacobi." IEEE Transactions on Automatic Control (2025).
[2] Zinage, Vrushabh, and Efstathios Bakolas. "Neural koopman lyapunov control." Neurocomputing 527 (2023): 174-183.
[3] Folkestad, Carl, Yuxiao Chen, Aaron D. Ames, and Joel W. Burdick. "Data-driven safety-critical control: Synthesizing control barrier functions with Koopman operators." IEEE Control Systems Letters 5, no. 6 (2020): 2012-2017.
[4] Lusch, Bethany, J. Nathan Kutz, and Steven L. Brunton. "Deep learning for universal linear embeddings of nonlinear dynamics." Nature communications 9, no. 1 (2018): 4950.

---

> ### Author Response · Authors · 2025-08-22
> **Rebuttal-1**
>
> We appreciate the reviewer’s feedback. Our responses to the specific questions and concerns are provided below.
>
>
> **Question 1.** `There have already been extensive works on the problem statement discussed in this paper. In addition, their algorithms have been applied to complex high dimensional robotic systems`
>
> We thank the reviewer's question. KEEC is not intended as an incremental tweak, but as a principled response to a fundamental open question in latent embeddings: _"what constitutes an appropriate embedding for control?"_.
>
>
> 1. **Regarding “extensive existing work”.** We contribute a principled framework defining two necessary conditions—equivariance and isometry—for control-consistent embeddings.
>
> While prior methods can be empirically effective, they typically emphasize next-state prediction and overlook two key aspects of dynamical systems: the equivariant flow/vector field and the metric structure. This omission can cause policies that are optimal in the latent space to fail when transferred to the original system.
>
> Our core innovation lies precisely here: Rather than embedding only the states, we embed the entire structure of dynamics. We show that maintaining **equivariance** and **isometry** is necessary for control consistency between the original and latent spaces.
>
> - **Embedding Part.** Technically, in our embedding, the Koopman embedding map $g$ induces the equivariance of flows and vector fields based on manifold and Lie theory (see Appendix D, pp. 20–21). By leveraging the infinitesimal generator (the Lie algebra), KEEC is able to comprehensively embed both equivariant flows and vector fields into the latent space. However, equivariance alone is insufficient; we therefore incorporate an **isometry** loss (as empirically validated in Figure 4). In control tasks, rewards/policies depend on the **intrinsic metric** (e.g., distances to targets). If the embedding distorts this metric, a policy optimal in the latent space may fail in the original space.
>
> - **Control Part.** Owing to the comprehensive embedding that satisfies both equivariance and isometry, we obtain an invariant value function (see Lemma 3.3), ensuring that control performance is consistent across the original and latent spaces. Building on this, the learned equivariant vector fields and invariant value function enable us to derive an analytical policy directly, without the need to train a separate policy neural network (see Theorem 3.4). This analytical policy can then be incorporated into Bellman value iteration, which not only provides a strong convergence guarantee toward the optimal value function (see Appendix F.3) but also largely accelerates policy generation (see left Figure 3).
>
> Thus, our novelty is not a new problem statement but a more rigorous, reliable solution to latent embedding that ensures control consistency.
>
>
> 2. **On the application to high-dimensional robotic systems.**
>
> We focus on diverse nonlinear dynamics (chaos, PDEs, visual input), which generalize beyond robotics and are equally challenging.
>
> - **Lorenz-63 system:** evaluates robustness under chaotic dynamics, where sensitivity to initial conditions makes purely local models ineffective. Our embedding directly addresses this challenge.
>
> - **Wave equation:** serves as a benchmark for controlling high-dimensional (50-dimensional) partial differential equation (PDE) systems, demonstrating scalability and generality beyond rigid-body dynamics.
>
> - **Image-based pendulum:** validates that KEEC can learn control policies directly from high-dimensional (4608-dimensional) visual inputs.
>
> We believe this clarifies that KEEC does not propose a new problem statement, but rather a principled and theoretically grounded solution. While we have not yet applied it to robotics systems, our current experiments demonstrate its generality across diverse and challenging dynamical systems, and robotics applications are a natural next step.

---

> ### Author Response · Authors · 2025-08-22
> **Rebuttal-2**
>
> **Question 2.** `This paper compares their algorithm with classical yet simpler nonlinear systems that have already been well studied in literature.`
>
>
> We thank the reviewer for the comment. While the benchmarks are classical, they were chosen precisely because they capture fundamental challenges—such as chaos, high-dimensional PDE dynamics, and raw visual control—where even strong baselines often fail. This demonstrates that the tasks are nontrivial and provide a meaningful testbed for our method.
>
> 1. **The baselines are competitive and widely recognized in the RL community.** We compare KEEC against a suite of competitive reinforcement learning and control algorithms, including Soft Actor-Critic (SAC, state-of-art online RL algorithm [5]), Conservative Q-Learning (CQL), Model Predictive Path Integral (MPPI) and Prediction, consistency, and Curvature (PCC). These baselines are representative and cover modern control and RL algorithms.
>
> 2) **Each system targets a distinct, nontrivial challenge (beyond mere dimensionality):**
> - **Lorenz-63 (chaotic dynamics).** A canonical chaotic system with sensitive dependence on initial conditions and nonlinearity that defeats purely local models; PCC, CQL and MPPI fail to stabilize/track reliably in our experiments.
> - **Wave equation (high-dimensional PDE control).** An infinite-dimensional system; our discretization yields a 50-dimensional state with its velocity. Despite this classical setup, strong baselines (e.g., SAC) fail to control the unknown dynamics, highlighting the task’s difficulty.
> - **Image-based pendulum (high-dimensional visual input).** Requires learning control directly from $96\times48$ raw images and acting in the learned embedding—standard yet challenging for methods that must couple representation learning with control.
>
> **Additional Result Analysis.** As shown in Table 1 and Figure 2, KEEC achieves substantially higher rewards and more robust trajectories than strong baselines such as SAC and PCC, particularly on chaotic (Lorenz-63) and high-dimensional PDE (Wave) systems. These results further confirm that the benchmarks are nontrivial and provide a meaningful testbed for evaluating control algorithms.
>
> While classical, these benchmarks are not “simple”: they stress chaos, high-dimensional PDE control, and high-dimensional visual inputs. This diversity provides a sharper testbed for our theoretical claims than narrowly focusing on robotics alone.

---

> > ### Author Response · Authors · 2025-08-22
> > **Rebuttal-3**
> >
> > **Question 3.** `Furthermore, the authors have not compared their approach with existing Koopman based control approaches (please see [1]. [2], [3], [4] and the references therein)`
> >
> > **Answer.** Thank you for pointing out these references. We have added an explicit comparison in the revised **Related Work**. In brief, KEEC differs from [1]–[4] in scope, problem settings, assumptions, and technical mechanism:
> >
> > 1. **Vaidya (2025): “When Koopman Meets Hamilton and Jacobi.”**
> >    - **Scope/background.** Establishes theoretical links between Koopman spectral theory and Hamiltonian/Lagrangian manifolds; it is not a deep-learning paper and proposes no concrete control embedding or control algorithm, so it is not a direct baseline for KEEC.
> >    - **Relation to KEEC.** The Conclusions [1] explicitly suggest data-driven extensions for control analysis/synthesis. KEEC advances this direction by:
> >
> >          (a) learning a Koopman embedding that enforces equivariance (flows & vector fields via the infinitesimal generator) and isometry (metric preservation),
> >          (b) constructing an invariant value function linked to the HJ equation (Section 3.4),
> >          (c) deriving an analytical policy without a separate policy network that plugs into Bellman value iteration with convergence guarantees.
> >
> > 2. **Zinage & Bakolas (2023): “Neural Koopman Lyapunov Control.”**
> >    - **Scope/background.** Learns a stabilizable bilinear Koopman model and a Control Lyapunov Function (CLF) via a learner–falsifier loop; the focus is **stabilization**.
> >    - **Assumptions/goal.** Assumes a **unique equilibrium** and emphasizes CLF-based stability in the lifted model—assumptions not suitable for systems like **Lorenz-63**  (chaotic, no unique equilibrium, see more details of Lorenz-63 in Appendix H.1).
> >    - **Mechanism vs KEEC.** Does not enforce **equivariance** of flows/vector fields nor **isometry**, and does not derive an **HJ-based analytical policy** on the invariant value function for reward maximization, as KEEC does.
> >
> > 3. **Folkestad et al. (2020): “Data-Driven Safety-Critical Control via Koopman CBFs.”**
> >    - **Scope/background.** Synthesizes **control barrier functions (CBFs)** for **safety** using Koopman operators; primary objective is safety guarantees.
> >    - **Assumptions/goal.** Relies on incremental stability and invariant set assumptions for dynamics (the goal in [3] is to ensure safety); this goal differs from KEEC’s optimal-control focus.
> >    - **Mechanism vs KEEC.** Works with CBF construction on (discrete-time) Koopman models. In contrast, KEEC emphasizes a **comprehensive embedding** of **flows and vector fields** with **equivariance + isometry** (see Proposition 3.2 and Sec. 3.2).
> >
> > 4. **Lusch, Kutz & Brunton (2018): “Deep Learning for Universal Linear Embeddings.”**
> >    - **Scope/background.** Learns Koopman eigenfunction representations (autoencoder-like) to obtain globally linear dynamics for **prediction/estimation**; it is **not** a control algorithm.
> >    - **Assumptions/goal.** Aims to discover coordinates/eigenfunctions; does **not** address control-consistency (no isometry) and thus the embedding does not directly support control.
> >    - **Mechanism vs KEEC.** Identifies Koopman coordinates (including continuous spectra) but does not treat **equivariant flows/vector fields**, **invariant value functions**, or **analytical policies** integrated with value iteration.
> >
> >
> > **KEEC in contrast.** For dynamics, we have less assumptions on its forms, stability or invariant sets. We target **embedding design for control**: we **learn** a Koopman embedding that enforces **equivariance** (flows & vector fields) and **isometry** (metric preservation), yielding an **invariant value function** (lemma 3.3) and an **HJ-consistent analytical policy** (Equation (16)) that integrates with **Bellman value iteration** (convergence guarantees of value iteration see Appendix F.3) **without** training a separate policy network. We have incorporated a concise comparison with [1]–[4] in the revised Related Work.

---

> ### Author Response · Authors · 2025-08-22
> **Rebuttal-4**
>
> **Question 4.** `The loss functions considered in this paper such as equations (8), (9), (10) and (11) have already been considered in several papers in the research literature`
>
>
> Thank you for this comment. While related ideas have been explored in prior work [1–4], **our loss formulations differ in several fundamental ways:**
>
> 1. **Regarding the Formulation (8).**
>    Our formulation for the actuation operator, $\mathcal{U} \in \mathbb{R}^{n \times n \times d}$, is a 3-mode tensor, where we explicitly models the state-dependent effects of $z$ on $\mathcal{U}$. In contrast, prior works [2–3] assume a state-independent control matrix during embedding, which is a simplification that we do not make.
>
> 2. **Regarding the Forward/Equivariance Loss (10).**
>    Unlike [2–4], our objective is to simultaneously embed both equivariant flows and vector fields which distinguishes our work from [2–4] where the goal is limited to embedding discrete-time states. We achieve this by first identifying the infinitesimal generator $[\mathcal{P}, \mathcal{U}]$ (the local Lie algebra) and then using it to construct our equivariant flow prediction (as shown in Proposition 3.2, $z_{\Delta t}\approx \exp(\mathcal{P}\Delta t) z_0 + \mathcal{P}^{-1} \left(\exp(\mathcal{P}\Delta t) - I\right)\mathcal{U}(z_0)a_0$). By contrast, the goal of [2–4] is limited to embedding discrete-time states, which is fundamentally different from our target. In our case, the formulation extends beyond discrete dynamics, enabling an embedding of continuous-time flows themselves, which provides a more faithful representation of system evolution.
>
> 3. **Regarding the Isometry Loss (11).**
>    We did not find an isometry loss in [1–4]. Our contribution emphasizes isometrically embedding the manifold of dynamics while minimizing metric distortion. To the best of our knowledge, this aspect has not been discussed in the cited works. This loss is crucial for ensuring that control effects in the latent space transfer faithfully back to the original space. For example, Figure 4 in our paper clearly shows that without the isometry embedding (Equation 11), the latent space becomes discontinuous and unfaithful. Similarly, as shown in the last two rows of Table 1, removing the isometry loss leads to substantially lower rewards compared to KEEC with Equation 11 included.
>
> We summarize the differences in Table a. While [1–4] primarily focus on stability, safety, or representation learning, our goal is to answer the question: “_what constitutes an appropriate embedding for control?_” To this end, KEEC introduces loss functions explicitly tailored for control-consistent embedding, jointly enforcing equivariance and isometry, and further supoorting the derivation of an analytical control algorithm with an invariant value function—an aspect not introduced in prior work.
>
>
> **Table a. Comparison of Loss Functions: Prior Work vs. KEEC**
>
> | **Loss** | **Prior Work [1–4]** | **KEEC** | **Key Difference** |
> |----------|----------------------|----------|---------------------|
> | **(8) Actuation matrix $\mathcal{U}$** | Assumes **state-independent** control matrix | Models $\mathcal{U}$ as a **state-dependent 3-mode tensor** | Captures richer, realistic state-dependent effects |
> | **(10) Forward / Equivariance** | Embeds **discrete-time states only** | Embeds **flows + vector fields** via infinitesimal generator $[\mathcal{P},\mathcal{U}]$ | Goes beyond discrete dynamics → continuous-time flows |
> | **(11) Isometry** | **Not present** | Enforces **metric preservation** to ensure consistent control performance in original and latent spaces | New contribution; Figure 4 & Table 1 show necessity |
>
>
>
> References
>
> [1] U. Vaidya, When Koopman Meets Hamilton and Jacobi, IEEE TAC, 2025.
>
> [2] V. Zinage & E. Bakolas, Neural Koopman Lyapunov Control, Neurocomputing, 2023. (github: https://github.com/Vrushabh27/Neural-Koopman-Lyapunov-Control/blob/main/Simple_Pendulum.ipynb)
>
> [3] C. Folkestad et al., Data-driven Safety-Critical Control: Synthesizing Control Barrier Functions with Koopman Operators, IEEE L-CSS, 2020.
>
> [4] B. Lusch, J. N. Kutz, S. L. Brunton, Deep Learning for Universal Linear Embeddings of Nonlinear Dynamics, Nat. Commun., 2018.
>
> [5] https://github.com/quantumiracle/Popular-RL-Algorithms
>
> -----------------------------
> We sincerely appreciate the reviewer’s comments, which have helped us improve the manuscript. We hope our clarifications resolve the concerns raised.

---

### Decision · Action_Editor_qEBF · 2025-09-27

**Recommendation:** Reject

**Additional Comments:**

Given the reviews and rebuttals, although the paper has its non-negligible merits, at this point I'm not confident enough to recommend acceptance since the central claims of this paper would require more thorough and in-depth justification. The final recommendation is therefore rejection with the possibility to resubmit. In terms of the revision, a suggestion is to refine the claims, concretely back each one up (particularly in the context of existing Koopman literature when arguing the novelty of equivariance and isometry), and restructure the paper around them if necessary.

**Audience:**

Yes

**Audience Explanation:**

Regarding the algorithmic contribution, two of the three reviewers recognized that the paper would be of interest to a part of the community. Although all reviewers expressed concerns on the significance of this paper, we are aware that significance is not within the key evaluation criteria of this journal.

Regarding the empirical contribution, as far as I can see the experiments in this paper are extensive and well executed. Therefore I believe the experiments would be of interest to the community as well.

**Claims And Evidence:**

No

**Claims Explanation:**

The submitted paper investigates the use of the Koopman operator theory to learn the latent embeddings of unknown nonlinear dynamical systems, as well as the subsequent use of this technique in downstream control tasks. Compared to a rich body of existing works on this topic, the paper claims its main contribution/novelty as showing the importance of two design objectives: equivariance and isometry. Following this idea, an algorithm is proposed by incorporating an additional equivariance loss and an isometry loss into the deep learning + Koopman pipeline. The proposed algorithm is compared to a number of non-Koopman baselines in extensive experiments.

The reviewers have varied opinions on this paper. Among the three final recommendations collected from the reviewers, there is one weak accept, one weak reject and one reject. A notable criticism is that the ideas of equivariance and isometry (more specifically, the corresponding losses Eq.10 and Eq.11) are not new for the Koopman literature, to which the submitted paper has not given sufficient credit. Quoting the one reviewer who has worked in this area before, they believe the paper "provides no new insights that were not present in previous research literature". If true, this would considerably weaken the validity of the paper's central claim.

On my end, I have carefully read the authors' response as well as relevant parts of the paper. Although I must admit my limited experience in Koopman operator theory, I do find the paper's coverage of existing works not sufficiently in-depth, and somewhat not exactly on target. To fully justify the scientific contributions of this paper, key questions are
1. whether equivariance and isometry have been considered in the Koopman literature before, and
2. if they have been considered before, what are the key difference and strength of the proposed technique.

The authors' response to Reviewer vvzJ partially answered these questions by comparing to the four papers the reviewer linked, but the broad picture remains incomplete to a large extent. Therefore I am leaning towards the reviewer's point and find the above a notable limitation of this paper.

**Resubmission Of Major Revision:**

The authors may consider submitting a major revision at a later time.